# Efficient and scalable synthesis of highly aligned and compact two-dimensional nanosheet films with record performances

Jing Zhong[1], Wei Sun[2,3], Qinwei Wei[1,4], Xitang Qian[1,4], Hui-Ming Cheng [1,4,5] & Wencai Ren [1,4]

It is crucial to align two-dimensional nanosheets to form a highly compact layered structure for many applications, such as electronics, optoelectronics, thermal management, energy storage, separation membranes, and composites. Here we show that continuous centrifugal casting is a universal, scalable and efficient method to produce highly aligned and compact two-dimensional nanosheets films with record performances. The synthesis mechanism, structure control and property dependence of alignment and compaction of the films are discussed. Significantly, 10-μm-thick graphene oxide films can be synthesized within 1 min, and scalable synthesis of meter-scale films is demonstrated. The reduced graphene oxide films show super-high strength (~660 MPa) and conductivity (~650 S cm$^{-1}$). The reduced graphene oxide/carbon nanotube hybrid-film-based all-solid-state flexible supercapacitors exhibit ultrahigh volumetric capacitance (407 F cm$^{-3}$) and energy density (~10 mWh cm$^{-3}$) comparable to that of thin-film lithium batteries. We also demonstrate the production of highly anisotropic graphene nanocomposites as well as aligned, compact films and vertical heterostructures of various nanosheets.

[1] Shenyang National Laboratory for Materials Science, Institute of Metal Research, Chinese Academy of Sciences, 72 Wenhua Road, Shenyang 110016, China. [2] School of Civil Engineering, Harbin Institute of Technology, Harbin 150090, China. [3] Key Lab of Structure Dynamic Behavior and Control (Harbin Institute of Technology), Ministry of Education, Harbin 150090, China. [4] School of Materials Science and Engineering, University of Science and Technology of China, 72 Wenhua Road, Shenyang 110016, China. [5] Tsinghua-Berkeley Shenzhen Institute (TBSI), Tsinghua University, 1001 Xueyuan Road, Shenzhen 518055, China. Correspondence and requests for materials should be addressed to W.R. (email: wcren@imr.ac.cn)

Atomically thin two-dimensional (2D) materials have attracted tremendous interests since the discovery of graphene because of the unusual electronic, mechanical and optical properties as well as a wide range of intriguing applications[1–4]. Most of the extraordinary properties of 2D materials originate from their unique in-plane atomic-bonding structure. For example, the in-plane $sp^2$ carbon–carbon covalent bonding of graphene leads to massless Dirac Fermions behavior of charge carriers as well as giant mobility, super-high mechanical strength and record thermal conductivity in the in-plane direction[1,2]. In contrast, the corresponding out-of-plane properties are generally significantly lower. Such intrinsically extremely anisotropic properties suggest that it is crucial to align the mass-produced 2D nanosheets into a highly compact layered film structure to enable their excellent in-plane properties being utilized to the maximum extent at macroscopic scale.

Currently, several strategies have been developed to synthesize 2D nanosheets films, such as vacuum filtration (VF)[5], interfacial assembly[6], Langmuir-Blodgett assembly[7], rod/dip/spray coating[8–10], and gel-film transformation (GFT)[11]. The films obtained show promising applications[5–25] in electronics, optoelectronics, thermal management, supercapacitors, lithium batteries, protective coatings, and molecular/ion separation. However, the alignment and compaction of 2D nanosheets in the films and the film properties are still limited. For instance, the highest strength reported for the common reduced graphene oxide (rGO) films is lower than ~300 MPa along with a small electrical conductivity of ~120 S cm$^{-1}$[22], and the volumetric energy density of 2D nanosheets-based thin-film supercapacitors is much lower than that of the thin-film lithium batteries[23–25]. Moreover, these methods are not suitable for scalable production in terms of either time consuming, complicated procedure, high cost, or specific requirements on the 2D nanosheets and their dispersion. For instance, it usually takes several days to fabricate a ~10-μm-thick graphene oxide (GO) films by the commonly used VF because of the greatly reduced water permeability with the thickness[5], and only GO nanosheets with low defect density are applicable for the GFT process along with a long synthesis time of over 80 h[11]. In addition, it remains a great challenge to achieve good alignment of 2D nanosheets in composites although the alignment is critically important to improve the reinforcement efficiency.

Here, we report a universal, highly efficient and scalable method, continuous centrifugal casting (CCC), to produce highly aligned and compact 2D nanosheet films with record performances in many applications. The physical mechanism and control strategies for the alignment and compaction of 2D nanosheets by CCC method are first analyzed with GO as a model system based on fluid mechanics. We then demonstrate the synthesis of highly aligned and compact GO films, super-strong and highly conductive rGO films, rGO/single-walled carbon nanotubes (SWCNTs) hybrid films for flexible supercapacitors (SCs) with record volumetric energy density, highly anisotropic graphene nanocomposites as well as various 2D nanosheet films and vertical heterostructures, by CCC method. The influence of alignment and compaction degree on the properties of films as well as the high efficiency, easy scalability and good controllability of CCC method are also demonstrated and discussed.

## Results

**CCC method and synthesis of GO films.** Taking GO nanosheets as an example, Fig. 1a shows the CCC production process of highly aligned and compact 2D nanosheet films. The GO nanosheets were synthesized by traditional Hummers method[26], which are all monolayers with a C/O ratio of 1.7 and average

lateral size of ~1 μm. If not specified, we used these GO nanosheets in our experiments. During CCC, the GO dispersion (8 mg mL$^{-1}$) was continuously casted/sprayed on the inner surface of a rotating hollow tube (RHT) accompanied with a low temperature heating (80 °C). On the one hand, the continuously high-speed rotating of the RHT induced strong centrifugal force along the radial direction of the RHT (Supplementary Fig. 1 and Supplementary Note 1). On the other hand, the velocity difference between the RHT and casted/sprayed liquid led to shear force along the tangential direction (Supplementary Fig. 1 and Supplementary Note 1). Figure 1b shows the simulated transient shear rate distribution at the beginning of the tube rotation (at 1 ms). It can be clearly seen that the momentum generated by tube rotation was immediately transported from the liquid–solid boundary (inner surface of RHT) into the GO dispersion because of the intrinsic viscosity of the GO dispersion, generating uniform shear rate around the RHT inner surface. Therefore, the GO nanosheets in the dispersion were subject to both centrifugal force and shear force simultaneously during CCC. The former can lead to the compaction of GO nanosheets along the radial direction of the RHT, while the latter can not only align but also smooth the GO nanosheets along the tangential direction of the RHT. Meanwhile, the low-temperature heating of the RHT accelerates water evaporation and consequently helps the alignment and compaction of the GO nanosheets. As a result, highly aligned and compact GO films were formed (Fig. 1c, d).

Keeping the liquid in laminar flow is the prerequisite to align the 2D nanosheets by CCC. Based on the fluid mechanics[27], there are mainly two forces that control the behavior of the flow in the steady state, which are inertial force and viscose force. The former can be considered as the dynamic sensitivity of flow, while the latter as the dissipation capability for the disturbance. Typically, the relative contribution of these two forces can be estimated by the Raynolds number, which is the ratio of inertial force to viscose force. In our study, the liquid layer around the inside surface of the RHT, typically considered as rimming flow, is a non-trivial example of a steady 2D viscous flow with a free surface[28,29]. The Raynolds number is expressed as: $Re \equiv \omega D^2/v$, where $\omega$, $D$, and $v$ are the angular velocity of RHT, and the average thickness and kinematic viscosity of the flow, respectively. When the Raynolds number is small, any disturbation, including the change of the tube rotation rate and any uncertainty of the flow boundary condition (such as the roughness of the inner surface of the hollow tube), can be dissipated and the flow exhibits characteristic of laminar flow. When the Raynolds number is large, any disturbation will be amplified and eventually results in the generation of eddies, and the flow exhibits characteristic of turbulence. Therefore, the flow should be controlled within the laminar flow in order to align the 2D nanosheets. Here, we take the critical Raynolds number to be one, to roughly estimate the domains corresponding to laminar flow phase and turbulent phase, and the phase diagram obtained is shown in Supplementary Fig. 2. It is surprising to see that the maximum rotating rate that is allowed to keep the solution in the laminar flow region should be no <$10^9$ rad s$^{-1}$, corresponding to tube rotating rate of ~$10^{10}$ r min$^{-1}$, considering the viscosity (~1.26 Pa s) of the GO dispersion used in our experiments. A much larger rotating rate is allowed if considering the increased viscosity caused by the water evaporation.

In order to achieve better alignment and compaction by CCC, the shear stress and centrifugal stress should be as high as possible in the premise of laminar flow. We further analyzed these two forces that are applied to GO nanosheets in the CCC process (Supplementary Fig. 1 and Supplementary Note 1). The derived shear stress ($\sigma_{shear}$) is expressed as: $\sigma_{shear} = \mu\,\omega\,\mathbf{e}_\theta$ where $\mu$, $\omega$, and

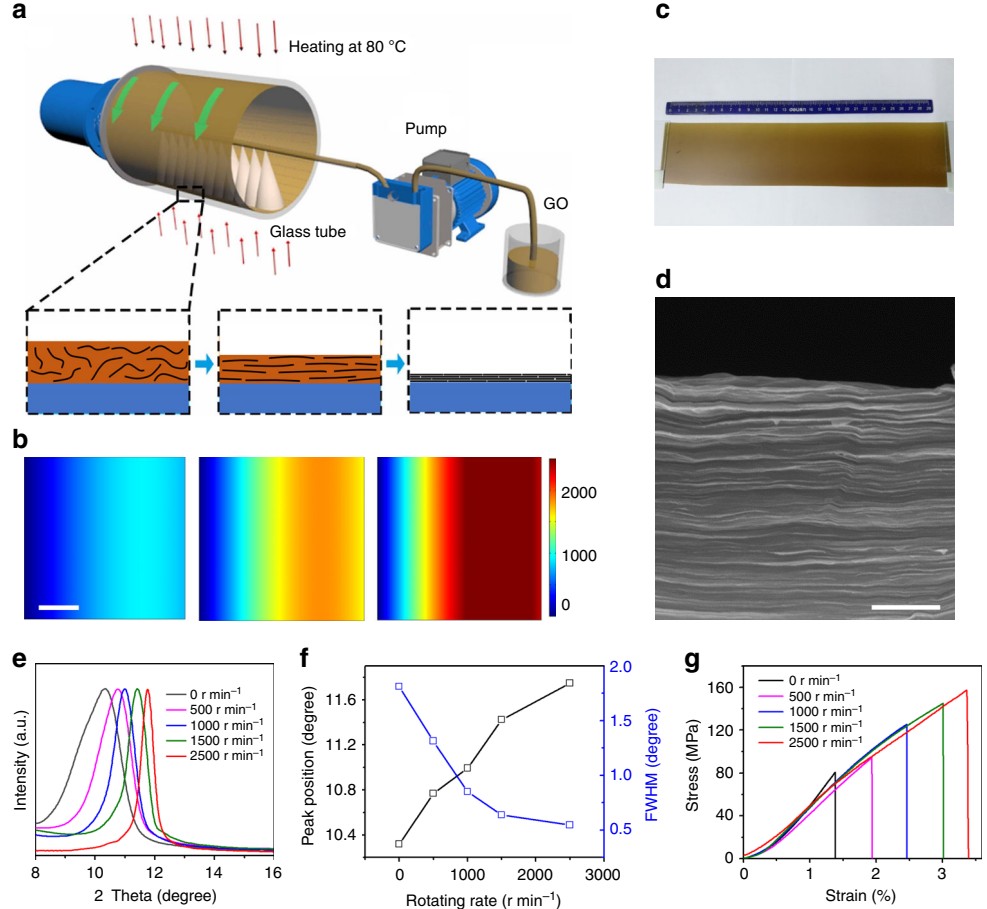

**Fig. 1** Production and characterization of highly aligned and compact GO films by CCC. **a** Schematic of the CCC production process. **b** Simulated transient shear rate field at different rotating rate. Note that the viscosity of the liquid greatly increases during the CCC process because of the evaporation of water by low-temperature heating. Therefore, we used a viscosity of 10 Pa s in the simulations, which corresponds to the concentration of GO dispersion of ~22 mg mL$^{-1}$. The horizontal axis from the right to the left represents the distance to the RHT wall, and the color bar shows the shear rate (s$^{-1}$). Left panel, 500 r min$^{-1}$; middle panel, 1000 r min$^{-1}$; right panel, 1500 r min$^{-1}$. **c** A GO film with a thickness of ~100 μm and size of ~30 × 10 cm$^2$. **d** Cross-sectional scanning electron microscopy (SEM) image of a GO film, showing highly aligned and compact layered structure. **e**–**g** XRD patterns (**e**) the corresponding XRD peak position and full width at the half maximum (FWHM) (**f**), and stress-strain curves (**g**) of the GO films produced with different rotating rate. Scale bars: **b** 50 μm; **d** 1 μm

$e_\theta$ are the viscosity of the liquid and the angular velocity and tangent unit vector of the RHT, respectively. The centrifugal stress ($\sigma_{centrifugal}$) is given by $\sigma_{centrifugal} = \rho_g\, t_g\, R_0\, \omega^2\, e_r$, where $\rho_g$, $t_g$, $R_0$, $\omega$, and $e_r$ are the density and thickness of GO nanosheets and the inner radius, angular velocity and radial unit vector of the RHT, respectively. It is clear that the shear stress increases linearly with the rotating rate of RHT and the viscosity of the liquid, and the centrifugal stress increases with the rotating rate and diameter of RHT. It has been reported that a shear stress of ~20 Pa is sufficient to align the GO nanosheets[30]. We estimated the shear stress applied to GO nanosheets in the GO dispersion used (8 mg mL$^{-1}$), which has a viscosity of 1.26 Pa s. The shear stress obtained is ~66, 131, 197, and 329 Pa for the rotating rate of 500, 1000, 1500, and 2500 r min$^{-1}$, respectively. Because of the flow is always in the region of laminar flow when the tube rotating rate is less than ~10$^{10}$ r min$^{-1}$, it is expected that the alignment and compaction of GO films can be improved by simply increasing the rotating rate from 0 to 2500 r min$^{-1}$. Note that the real shear stress should be much larger for each case during CCC process, since the viscosity of the liquid is increased with the evaporation of water by heating. Therefore, good alignment and compaction can also be achieved by using GO dispersion with much smaller concentration as raw material.

For the assembled films, the structure of 2D nanosheets has an important influence on their properties. It has been reported that the mechanical strength and electrical conductivity of the assembled GO and rGO films can be significantly improved by using GO nanosheets with low defect density and large lateral size[11,31]. More importantly, the properties strongly depend on the contact and interactions between stacked adjacent nanosheets since electrons/stress need to transfer between layers in the assembled films[5,22,31]. Considering the CCC process, the adjacent 2D nanosheets in the films are attracted each other mainly by Van der Waal's force. The van der Waals force can be estimated by Lifshitz's formular $F = S\frac{A_{Ham}}{6\pi d^3}$, where $A_{Ham}$ is the Hamaker coefficient, and $d$ and $S$ are the interlayer distance and contact area of two nanosheets, respectively[32]. It can be seen that the binding force is highly sensitive to the interlayer distance and proportional to the contact area between adjacent 2D nanosheets. Therefore, the contact and van der Waals force between stacked adjacent 2D nanosheets should be greatly increased with improving their alignment and compaction, and consequently improve the strength and electrical conductivity of the assembled films.

We then studied the structure and properties of the GO films synthesized at different rotating rate. X-ray diffraction (XRD) has

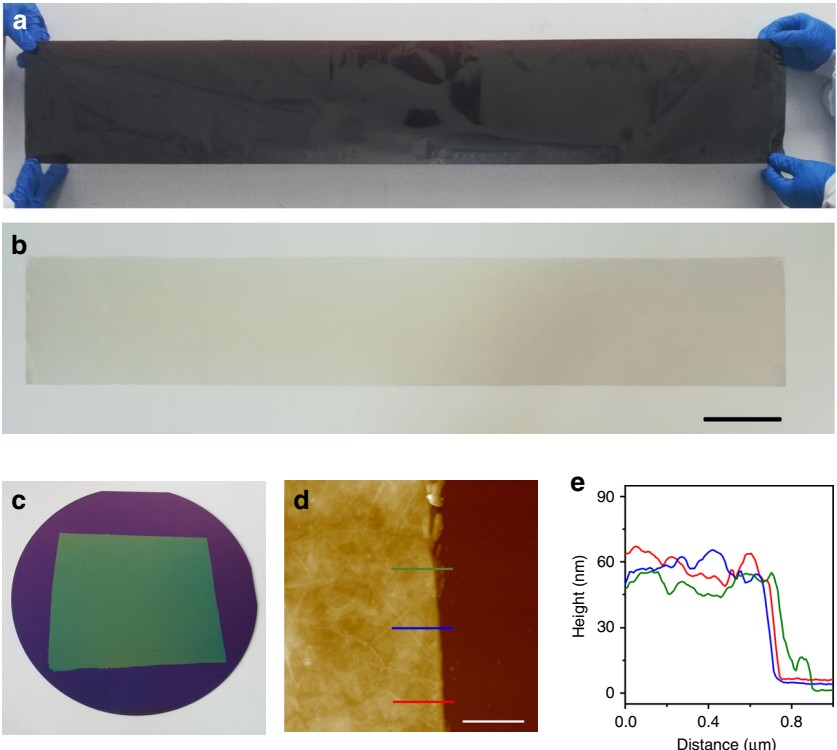

**Fig. 2** Scalable production of large-area highly aligned and compact GO films with different thickness. **a**, **b** A meter-scale ~10 μm-thick free-standing GO film (**a**) and thin GO film on polyethylene terephthalate (PET) (**b**) produced with a RHT of 33 cm in inner diameter. **c** A thin GO film on a 4 inch silicon wafer prepared with the same conditions as those for the production of the film in **b**. **d**, **e** Atomic force microscopy (AFM) image (**d**) of the film in **c** and the height profiles (**e**) along the three lines in **d**, showing the thickness of ~50 nm. Scale bars: **b** 10 cm; **d** 1 μm

been widely used to characterize the structure of assembled graphene- and GO-based films. Based on the Bragg's law, $2d\sin\theta = \lambda$, the position of XRD peak ($\theta$) determines the interlayer distance ($d$) of the films, and the FWHM reflects the ordering, which is correlated with the alignment and flatness of the nanosheets[5,20,22,33,34]. Figure 1e shows the XRD patterns of the GO films synthesized at different rotating rate. It can be seen that the XRD peak position of the GO films upshifts from 10.32° to 11.75° and the FWHM dramatically decreases from 1.81° to 0.55° when the rotating rate is increased from 0 to 2500 r min$^{-1}$ (Fig. 1e, f), confirming that the GO nanosheets become more compact and aligned. As discussed above, the higher compaction and better alignment can lead to better connection and larger van der Waals force between the stacked adjacent GO nanosheets. As a result, the tensile strength of GO films increases from ~80 MPa to ~157 MPa along with an increase in fracture strain from ~1.4% to ~3.4% (Fig. 1g). It is worth noting that the GO films obtained at 2500 r min$^{-1}$ shows smaller interlayer spacing (~0.75 nm) and FWHM (0.55°) than those of the reported GO films synthesized by other methods, typically in the range of ~0.8–0.95 nm and ~0.7–2.0°, respectively (Supplementary Table 1 and Supplementary Note 2), indicating better alignment and higher compaction. The much higher compaction and better alignment enable our GO films the highest tensile strength and fracture strain simultaneously among all the reported GO films made by GO nanosheets with similar structure even though the lateral size is much smaller. It is expected that the alignment and compaction of GO films and their mechanical properties could be further improved by increasing the rotating rate.

Furthermore, it is important to note that CCC allows us to continuously exert both high shear force and high centrifugal force along with the casting/spraying of GO dispersion, which enables highly efficient and scalable synthesis of highly aligned

and compact GO films. Typically, a 10-μm-thick film can be produced within 1 min by CCC. In contrast, it usually takes several days to fabricate a ~10-μm-thick GO film by VF, in which the GO nanosheets are assembled by directional water flow[5], because of the greatly decreased water permeability with the thickness[21,33], and hours by interfacial assembly[6], Langmuir-Blodgett assembly[7], and GFT[11]. Second, the thickness of the GO films can be easily tuned by simply changing the casting/spraying time and the concentration of GO dispersion. A dilute GO dispersion is usually beneficial for the production of thin films. Figures 1c and 2a–c show GO films with thickness of ~100 μm, 10 μm, 50 nm, and 50 nm, respectively. It can be seen that the GO films have uniform thickness and smooth surface (Fig. 2d, e). Note that it is almost impossible to synthesize very thick GO films by the commonly used VF. Third, the size of GO films can be easily scaled up by using a large RHT. For instance, we have synthesized meter-scale thick and thin GO films by simply using a RHT with a diameter of 33 cm (Fig. 2a, b). To the best of our knowledge, no such large highly aligned and compact GO films have been achieved by the present methods.

**Super-strong and highly conductive rGO films**. In order to obtain highly conductive graphene films, we reduced the highly aligned and compact GO films by using HI acid solution followed by thorough washing with de-ionized water and ethanol[35]. As shown in Supplementary Fig. 3, the resultant hydrophobic rGO films can be easily separated from the hydrophilic RHT by simply immersing into water. They are very flexible and can be folded into an origami crane without any damage (Fig. 3a, b). Figure 3c shows that the rGO sheets keep good alignment and compaction in the films. Based on the XRD results (Figs. 1e, f and 3d, e), the interlayer distance is greatly decreased from 0.75–0.86 nm to

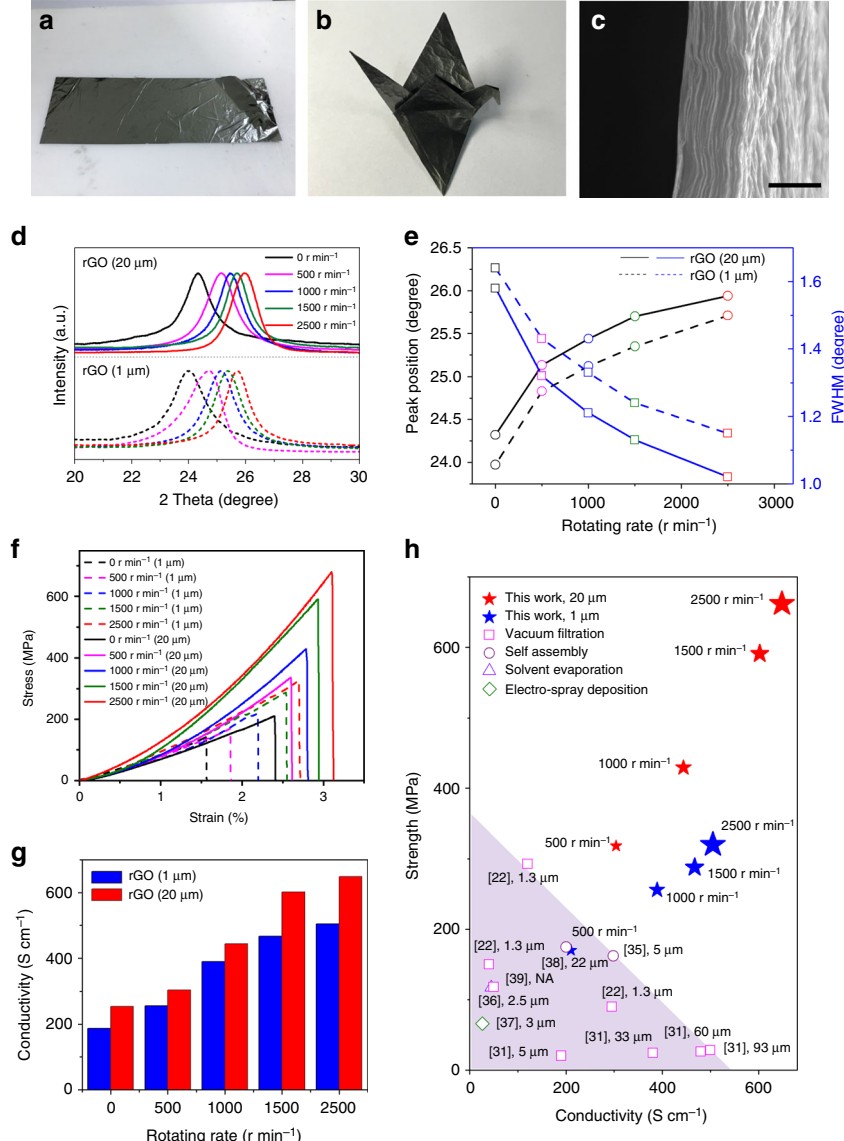

**Fig. 3** Structure, electrical and mechanical properties of rGO films. **a** A large rGO film with a size of 30 × 10 cm². **b** A origami crane made by rGO film, showing very good flexibility. **c** Cross-sectional SEM image of a rGO film. Scale bar: 1 μm. **d–g** XRD patterns (**d**), the corresponding XRD peak position and FWHM (**e**), stress-strain curves (**f**) and electrical conductivities (**g**) of the rGO films prepared at different rotating rate using GO nanosheets with different lateral size (~1 μm and ~20 μm). **h** Electrical conductivity and tensile strength comparison of the rGO films prepared by CCC method and other methods[22,31,35–39]. The size of the GO nanosheets used is shown for each case

0.35–0.37 nm after HI reduction. As a result, the tensile strength of rGO films is increased by ~70%–106% compared to GO films (Fig. 3f). Moreover, as the rotating rate increases, the interlayer distance decreases and alignment becomes better (Fig. 3d, e), and consequently the tensile strength and electrical conductivity increase (Fig. 3f, g). When the rotating rate is increased from 0 to 2500 r min⁻¹, the tensile strength and electrical conductivity of rGO films are increased by ~130 and 170%, respectively (Fig. 3f, g).

Moreover, the alignment, compaction, mechanical strength and electrical conductivity of the rGO films can be further increased by using large size GO nanosheets (Fig. 3d–g). Note that the rGO films obtained at 2500 r min⁻¹ with ~20-μm-large GO nanosheets have much smaller interlayer spacing (~0.34 nm) and FWHM (1.02°) simultaneously than those of the reported rGO films synthesized by other methods[22,31,35–39] (Fig. 3d, e and Supplementary Table 2). Together with the reduction in inter-

sheet contact, these films show super-high tensile strength of ~660 MPa and electrical conductivity of ~650 S cm⁻¹ at the same time (Fig. 3f, g). In contrast, the highest strength reported for the rGO films synthesized by GO nanosheets with similar chemical composition and structure, including the conjugated cross-linked films, is lower than 300 MPa along with a small conductivity of ~120 S cm⁻¹, and the highest conductivity is lower than 500 S cm⁻¹ along with a small tensile strength of 30 MPa although the size of the GO used is larger than 90 μm[22,31,35–39] (Fig. 3h and Supplementary Table 2). It is worth noting that the tensile strength of our films is also higher than that of AISI 304 stainless steel (585 MPa) and outperforms that (614 MPa) of the films synthesized with GO nanosheets with low defect density by GFT approach[11]. These results confirm that centrifugal casting is a very efficient method to fabricate highly compact and well-aligned graphene films with excellent combined properties.

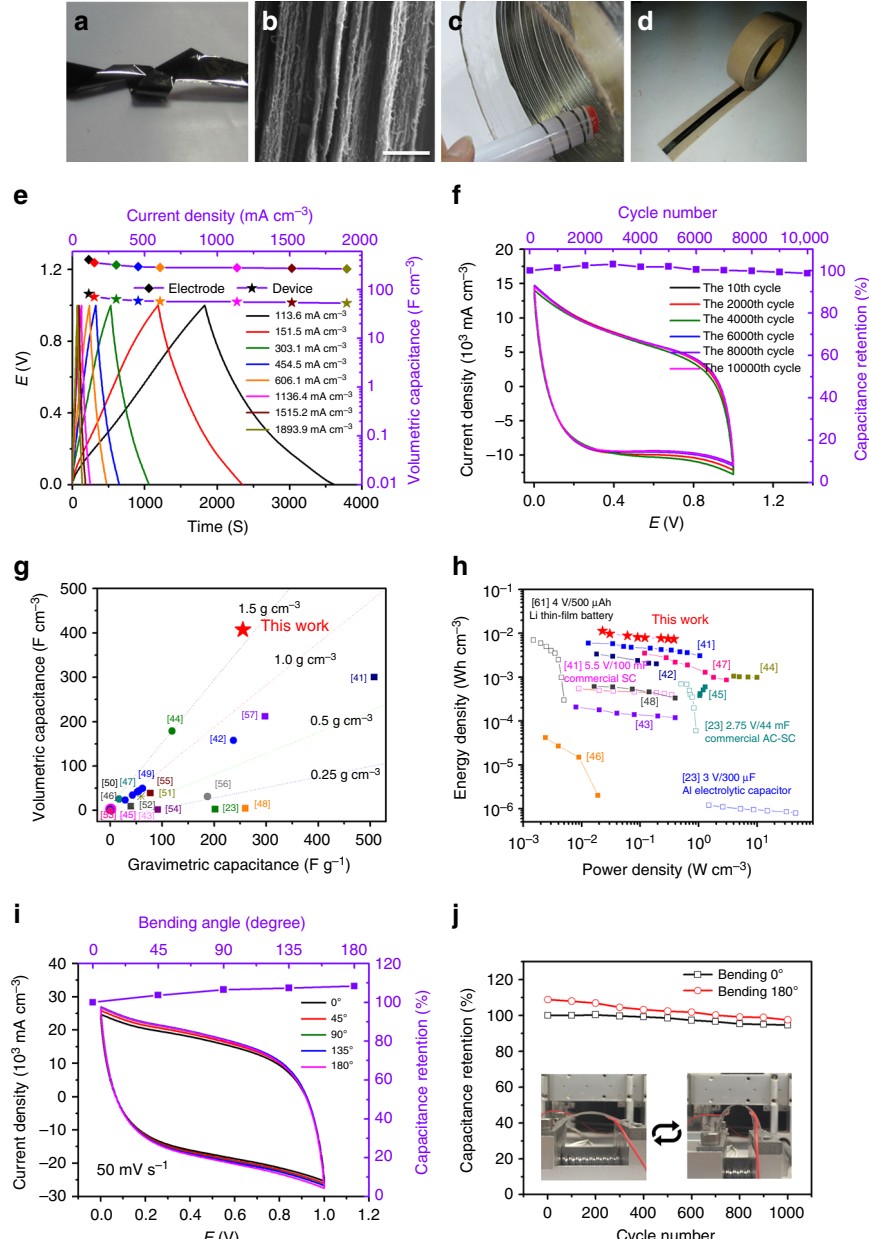

**Fig. 4** Structure and electrochemical performances of rGO/SWCNT hybrid-film-based flexible tape SCs. **a** A folded rGO/SWCNT hybrid film. **b** Cross-sectional SEM image of a rGO/SWCNT film. Scale bar: 1 μm. **c** Continuous long rGO/SWCNT hybrid ribbon obtained by cutting a film. **d** Tape SC made by the ribbon in **c**. **e** Galvanostatic charge/discharge curves measured between 0 and 1 V at current density from 114 to 1894 mA cm$^{-3}$, and the electrode and device volumetric capacitance as a function of current density. **f** Cycle life of the tape SC. CV curves after 10, 2000, 4000, 6000, 8000, and 10,000 cycles between 0 and 1 V at 100 mV s$^{-1}$, and the volumetric capacitance retention as a function of cycle number. **g** Volumetric and gravimetric capacitance comparison of rGO/SWCNT-1 hybrid film and the reported electrode materials. **h** Volumetric energy and power densities of our tape SC (red stars) compared with the reported all-solid-state SCs with different electrode materials (solid squares) and commercially available state-of-the-art energy storage systems (open squares). **i** CV curves collected at a scan rate of 50 mV s$^{-1}$ and the capacitance as a function of bending angle. **j** Capacitance retention under both flat and 180° bent states after 1000 cycles. Inset: a digital photograph showing the flexibility of the device

**rGO/SWCNT hybrid films for high-performance SCs.** Beside pure GO and rGO films, CCC method can also be used to produce highly compact and well-aligned hybrid films, for example, rGO/SWCNT films, by using a mixture dispersion of GO and SWCNTs as the raw materials. After CCC synthesis, the hybrid films were thoroughly washed with de-ionized water and ethanol to remove the sodium dodecyl sulfate (SDS), HI acid and iodine (Supplementary Fig. 4), which were used to disperse SWCNTs and reduce GO, respectively. As show in Fig. 4a, the rGO/SWCNT hybrid film obtained is also very flexible and can be

folded into complex structure without damage. SEM measurements show that the rGO sheets are well aligned and highly compacted with SWCNTs spaced between them with a very good contact (Fig. 4b). The SWCNTs not only act as spacers to prevent the stacking of rGO nanosheets to ensure a high specific area but also provide an electron transport highway together with the closely contacted rGO nanosheets. As shown in Table 1, with increasing the content of SWCNTs in the hybrid film, the density, tensile strength, and conductivity are decreased, but the specific surface area is increased. The hybrid films obtained with a mass

**Table 1 Properties of rGO/SWCNT hybrid films synthesized with different mass ratio of GO to SWCNT**

| Sample | GO:SWCNT mass ratio | Density (g cm$^{-3}$) | Tensile strength (MPa) | Conductivity (S cm$^{-1}$) | Specific surface area (m$^2$ g$^{-1}$) | Volumetric capacitance (F cm$^{-3}$) | Gravimetric capacitance (F g$^{-1}$) |
|---|---|---|---|---|---|---|---|
| rGO | — | 2.05 | 296 | 467 | <1 | 163.1 | 79.4 |
| rGO/SWCNT-4 | 4:1 | 1.79 | 121 | 231 | <1 | 210 | 117 |
| rGO/SWCNT-2 | 2:1 | 1.67 | 94 | 203 | 46 | 267 | 159 |
| rGO/SWCNT-1 | 1:1 | 1.59 | 71 | 184 | 129 | 407 | 255 |

ratio of GO to SWCNTs of 1:1 (rGO/SWCNT-1) show combined excellent properties that are desired for SCs: a high density of 1.59 g cm$^{-3}$, tensile strength of 71 MPa, electrical conductivity of 184 S cm$^{-1}$, and specific surface area of 129 m$^2$ g$^{-1}$.

It is well known that SCs store and release electrical energy based on the electrostatic interactions between ions in the electrolyte and electrodes near the electrode surface. Rational design of electrodes with developed ion/electron conductivity, enriched nanopores and high compaction are the key for improving the comprehensive performances of SCs. At micro-scale, the well-aligned rGO nanosheets and SWCNTs in our hybrid film form parallel oriented interlayer galleries, which can effectively restrict the electrolyte in such 2D space and consequently facilitate the transport of ions. More importantly, the high specific surface area and the compact layered structure of rGO/SWCNT electrodes not only allow the penetration of adequate amount of electrolyte but also greatly reduce the amount of in-effective electrolyte in the interlayer galleries, which ensure high volumetric and gravimetric capacitances for the whole device. At macroscale, because of the planar electrodes layout, ions can transport fast between the two electrodes reversibly without through considerable tortuosity as the case of fiber electrodes. In addition, the high electrical conductivity can facilitate rapid electron transport in electrodes, and the good flexibility and high tensile strength enable the use as flexible electrodes. Therefore, the rGO/SWCNT hybrid film is expected to be an ideal electrode material for high-rate flexible SCs with high capacitances.

Interestingly, the films produced by CCC method can be easily cut into continuous and long ribbons with a designed width by a sharp knife moving forward along the central axis of the rotating RHT controlled by a linear step motor (Fig. 4c). For instance, a meters-long ribbon with a width of 2 mm can be produced by our CCC equipment. Two identical hybrid ribbons were then adhered in parallel on a paper tape, infiltrated with the gel electrolyte of polyvinyl alcohol (PVA)/H$_2$SO$_4$, and finally sealed with poly (dimethyl siloxane) (PDMS), forming all-solid-state tape SCs (Fig. 4d). Usually, a small loading of the electrode material (a small thickness) results in high capacitance values. In order to avoid such exaggeration, it was suggested that the thickness of the electrode is preferred to be larger than 10 μm[40], which is comparable to that of the commercially available supercapacitors. In our experiments, the thickness of the electrodes is ~39 μm, and the width, length and mass are 2 mm, 50 mm, and ~6.2 mg, respectively.

Cyclic voltammetry (CV) measurements (Fig. 4e) show that the rGO/SWCNT-1 exhibits a super-high volumetric capacitance of 407 F cm$^{-3}$ at a current density of 114 mA cm$^{-3}$, and remain 264 F cm$^{-3}$ even at a very high current density of 1894 mA cm$^{-3}$. The Coulombic efficiency can reach 98%. Furthermore, our material has excellent cycling stability. As shown in Fig. 4f, the CV curves in 10,000 cycles almost overlap to each other, and the capacitance retention is ~99% after 10,000 cycles. It is worth noting that the volumetric capacitance of our hybrid films outperforms all the carbon-based electrode materials for all-solid-state SCs even those

hybrided with high-capacitance metal oxides and doped carbon materials[23,41–57] (Fig. 4g and Supplementary Table 3), which is about 1.4 times larger than the best value reported. Meanwhile, the gravimetric capacitance is comparable to the best value reported for the pure carbon-based electrodes[57]. The observed resistive CV curves might be due to the specific electrode layout of tape SCs and the resultant large electrode resistance, similar to the reported fiber SCs[41]. At the tens of micrometer scale, the thickness of the electrode has a small influence on the performance of the SCs. For example, reducing the thickness of electrodes to ~23 and 12 μm only leads to a small increase in volumetric capacitance to 419 and 433 F cm$^{-3}$, respectively, at a current density of ~110 mA cm$^{-3}$. MXenes is a new class of 2D materials with many intriguing properties such as good dispersion in water, excellent electrical conductivity and high volumetric capacitance[4,25,58,59]. For instance, the free-standing Ti$_3$C$_2$T$_x$ paper electrodes show a volumetric capacitance exceeding 900 F cm$^{-3}$ in 1M H$_2$SO$_4$[59]. It is reasonable to expect that highly aligned and compact MXenes papers can be synthesized by our CCC method to realize a super-high volumetric capacitance.

In order to further evaluate the energy storage performance of our SC, we calculated the volumetric power/energy density of the whole device. As shown in Fig. 4h, our SC has a volumetric energy density of ~9.98 mWh cm$^{-3}$, to our knowledge, which outperforms all the previously reported all-solid-state SCs based on different electrode materials, including hierarchically structured nitrogen-doped rGO/SWCNT hybrid fibers (6.3 mWh cm$^{-3}$)[41], graphene/CNT core-shell fibers (3.5 mWh cm$^{-3}$)[42], and MnO$_2$/carbon fibers (0.22 mWh cm$^{-3}$)[43], and over ~16 times higher than those of commercially available SCs[23,41,60]. More surprisingly, this energy density value is comparable to that of 4 V/500 μAh thin-film lithium battery (~0.3–10 mWh cm$^{-3}$)[61]. Meanwhile, our SC shows a superior power density with a maximum of ~380 mW cm$^{-3}$, which is higher than those of the commercially available 5.5 V/100 mF SCs[41] and more than 2 orders of magnitude higher than that of the thin-film batteries[61]. Furthermore, our SC shows a small change in capacitance under bending (Fig. 4i). It retains almost 90% of the initial capacitance both under flat and 180° bent states even after 1000 times of bending (Fig. 4j), demonstrating the high flexibility and excellent electrochemical stability of SCs.

For practical applications, SCs are usually connected either in series or in parallel to adjust the voltage window or discharge duration to meet the specific energy and power requirements. To demonstrate the practical applications of our SCs, three tape SCs with same length were assembled both in series and in parallel (Fig. 5). It can be seen from Fig. 5b, c, e, f, the charge/discharge and CV curves for the three single tape SCs in each case are almost overlapped, indicating the good uniformity of the ribbons. Connecting three tape SCs in parallel increases the output current and discharge time by a factor of three, at the same current density (Fig. 5a–c). For the three tape SCs connected in series, the charge/discharge voltage window increases from 1 V for a single tape SC to 3 V, with similar discharge time (Fig. 5d–f). These results show the great potential of our highly compact

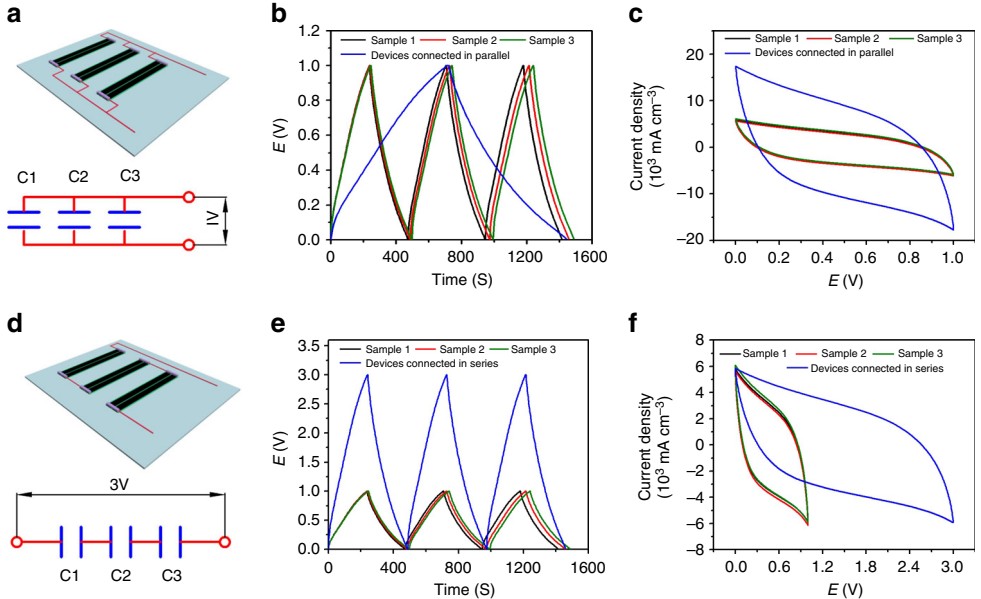

**Fig. 5** Assembly and electrochemical performance of multiple SC tapes. **a–c** Schematic and equivalent circuit (**a**), galvanostatic charge/discharge curves (**b**), and CV curves (**c**) of the three tape SCs connected in parallel. **d–f** Schematic and equivalent circuit (**d**), galvanostatic charge/discharge curves (**e**), and CV curves (**f**) of the three tape SCs connected in series. The galvanostatic charge/discharge curves and CV curves of the three corresponding individual tape SCs used to assemble multiple SCs were also presented

and well-aligned rGO/SWCNT hybrid-film-based tape SCs for practical energy storage applications.

**Highly anisotropic GO and rGO nanocomposites**. The CCC method also enables the scalable synthesis of highly anisotropic graphene/polymer nanocomposites (Fig. 6a). Figure 6b shows the electrical conductivity of millimeter-thick rGO/waterborne poly-urethane (PU) nanocomposites synthesized at different rotating rate. It can be found that, with increasing the rotating rate, the plane direction electrical conductivity significantly increases while the thickness direction conductivity dramatically decreases at the same time, indicating the development of graphene alignment. The anisotropic polymer obtained with 1500 r min$^{-1}$ shows 100 times difference in the conductivities along in-plane and thickness direction. Moreover, the Young's modulus are dramatically increased with increasing the alignment of GO sheets, although the tensile strength just shows a small increase (Fig. 6c, d). For instance, the Young's modulus of the nanocomposite prepared with 3 wt% GO at 1500 r min$^{-1}$ is increased by ~230% compared to the isotropic polymer at the same GO content and ~5900% compared to pure PU films. As shown in the inset of Fig. 6c, it can be further increased by over 5 times when the content of GO in the composite is increased from 3 wt% to 7 wt%.

**2D nanosheet films and vertical heterostructures**. Besides graphene-based nanosheets, the CCC provides a general strategy to synthesize highly compact and aligned films and vertical het-erostructures with essentially arbitrary 2D nanosheets. Supple-mentary Figure 5a–d shows various 2D nanosheet films on PET substrate synthesized by CCC method, ranging from insulator (hexagonal boron nitride, h-BN), semiconductor (black phos-phorus and WSe$_2$) to conductor (graphene). As an example, we also synthesized a vertical graphene/h-BN/WSe$_2$ heterostructure by sequentially casting graphene, h-BN, and WSe$_2$ dispersion on PET substrate. It can be seen that all the 2D nanosheets are highly compacted and well aligned along plane direction (Supplementary Fig. 5e). Furthermore, we can easily fabricate patterned 2D

nanosheet films by simply using a template covering the RHT (Supplementary Fig. 5f). These results demonstrate the great potential of our method for the fabrication of large-area 2D nanosheets-based electronics and optoelectronics devices.

## Discussion

We develop a novel CCC method to produce highly aligned and compact 2D nanosheet films. Fluid mechanics analyses indicate that the shear force and centrifugal force generated simulta-neously during CCC are responsible for the alignment and compaction of 2D nanosheets, respectively, and both forces can be dramatically improved by simply increasing the rotating rate of the rotating tube. It is found that the performances of the films strongly depend on the alignment and compaction degree. The super-high alignment and compaction of 2D nanosheets endow the films with excellent performances in many applications, such as super-strong and highly conductive rGO films as well as rGO/SWCNTs hybrid-film-based all-solid-state SCs with volumetric energy density comparable to that of thin-film lithium batteries. Compared to the present methods, this method also has advan-tages of high efficiency, easy scalability and good controllability in the structure of the films because of the continuous supply of both high shear force and high centrifugal force during the synthesis process. Furthermore, the CCC method is very uni-versal. As we demonstrated, it can also be used to synthesize highly anisotropic graphene nanocomposites as well as highly aligned and compact films and vertical heterostructures of various nanosheets.

With the rapid development of synthesis techniques[3,4,62,63], graphene and GO nanosheets with tunable chemistry and con-trolled structure and more and more 2D nanosheets with diverse properties have been produced. On the one hand, the good structure controllability of CCC opens up the possibilities to systematically investigate the influence of alignment and com-paction of 2D nanosheets on the various properties of the films. On the other hand, the CCC method will greatly promote the applications of 2D nanosheets in a broad range of areas such as

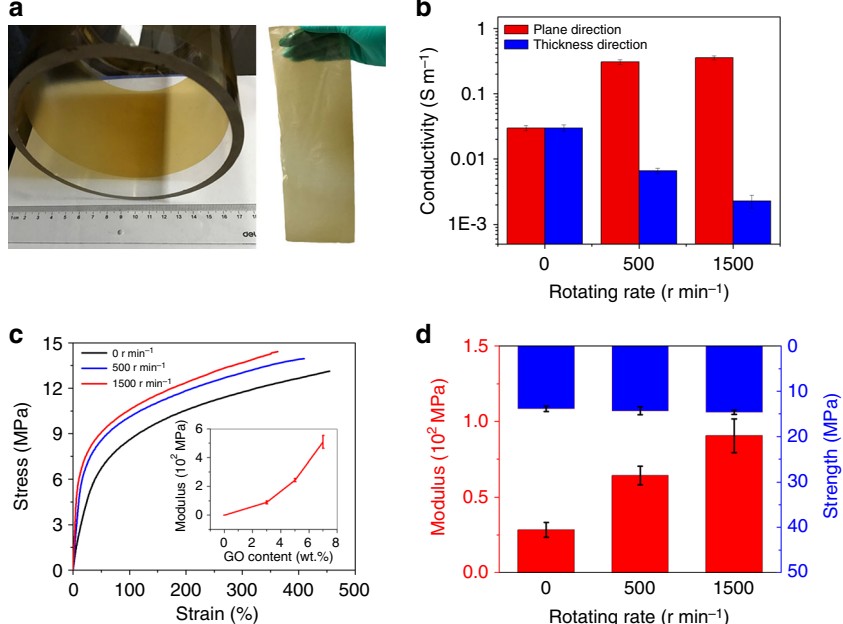

**Fig. 6** Structure, electrical, and mechanical properties of highly anisotropic GO (rGO)/PU nanocomposites. **a** A uniform rGO/PU nanocomposite film synthesized on the inner surface of RHT (left panel), which can be easily peeled off. A large rGO/PU film with a thickness of ~100 μm and size of 30 × 10 cm² (right panel). **b** Plane and thickness direction electrical conductivity of the rGO/PU nanocomposites films synthesized at different rotating rate. **c**, **d** Typical stress-strain curves (**c**), and strength and modulus (**d**) of the GO/PU nanocomposites films synthesized at different rotating rate. The content of rGO and GO nanosheets is 5 wt% and 3 wt% for the nanocomposites used for electrical conductivity and mechanical properties measurements, respectively. Inset of **c**, modulus of the GO films synthesized with a rotating rate of 1500 r min⁻¹ as a function of GO content

electronics, optoelectronics, thermal management, energy storage, membrane technology, protective coatings, and functional nanocomposites. In particular, it offers great flexibility in the fabrication of large-area vertical heterostructure films with specific properties by casting/spraying different 2D nanosheets in a precisely chosen sequence.

## Methods

**Synthesis of GO, rGO, and rGO/SWCNT films by CCC**. Natural graphite flakes (325 and 100 mesh, purity 99.5%) were purchased from Sigma-Aldrich. Small GO nanosheets (average size of ~1 μm) were synthesized using small graphite flakes by Hummers method[26]. Large GO nanosheets (average size of ~20 μm) were synthesized using large graphite flakes as reported in ref. [31]. SWCNTs (Elicarb® SW, diameter ~2 nm, length ~1 μm, and purity >90%) were purchased from Thomas Swan, UK.

GO films were synthesized by continuously spraying or casting GO dispersion (8 mg mL⁻¹) onto the inner wall of a heated rotating silica tube (inner diameter, 110 mm). The rotating rate of the silica tube was 500, 1000, 1500, and 2500 r min⁻¹, and the heating temperature was 80 °C. To synthesize ~50-nm-thick thin GO films, a very dilute GO dispersion with a concentration of 0.05 mg mL⁻¹ was used. After the GO films reached the designed thickness and the water was completely evaporated, the rotating was stopped and the films were directly peeled off from the silica tube. To synthesize rGO films, HI acid solution with a concentration of 10 wt % was sprayed on the surface of GO films at 120 °C. After cooling to room temperature, the films were thoroughly washed by de-ionized water (18 h) and ethanol (2 h), and then immersed into water and peeled off from the silica tube.

To synthesized rGO/SWCNT hybrid films, SWCNTs (5 wt%) was first ultrasonic dispersed in sodium dodecyl sulfate (SDS) solution (1.2 wt%) by ultrasonic cell pulverizer (SCIENTZ, JY 92-IIN) for 750 min with a power of 48 W. Homogeneous SWCNT/GO dispersions were prepared by mixing SWCNT dispersion with GO dispersion (5 mg mL⁻¹) with different SWCNT/GO mass ratio of 1:1, 1:2, and 1:4. Similar to the synthesis of GO films, the dispersion was then sprayed onto the inner wall of a heated rotating silica tube (inner diameter, 110 mm). The rotating rate of the silica tube was 1500 r min⁻¹, and the heating temperature was 80 °C. After the complete evaporation of water, HI acid solution with concentration of 10 wt% was then sprayed on the surface of the obtained GO/ SWCNT hybrid films at 120 °C to reduce them to rGO/SWCNT. After cooling to room temperature, the rGO/SWCNT films were peeled off from the silica tube, and thoroughly washed with de-ionized water (18 h) and ethanol (2 h) in sequence to get rid of SDS, HI acid and iodine. The rGO/SWCNT hybrid films synthesized with

SWCNT/GO mass ratio of 1:1, 1:2, and 1:4 were named as rGO/SWCNT-1, rGO/ SWCNT-2, and rGO/SWCNT-4, respectively.

**Synthesis of GO/PU and rGO/PU nanocomposites**. Waterborne polyurethane (PU, ~30 wt.%, Bye Gemany) was first mixed with GO dispersion (8 mg mL⁻¹) via tip sonication. Then the mixture was casted on the inner surface of a rotating silica tube heated at 80 °C. A GO/PU composite film was formed after the evaporation of water, and directly peeled off from the silica tube. To obtain rGO/PU nano-composite, the as-synthesized GO/PU composite was heated at 200 °C under ambient conditions for 2 h. The geometrical size of the samples for mechanical and electrical properties testing was 10 (length) × 5 (width) × 0.6 (thickness) mm³.

**Synthesis of 2D nanosheet films and heterostructures**. All the 2D nanosheets were exfoliated from their "mother" materials following the procedure reported by Coleman's group[64]. WSe₂, black phosphorous, h-BN powders were purchased from XFNANO and used as supplied. Each powder (1 mg mL⁻¹) was sonicated in Dimethylformamide by sonic bath (Branson 2510E-MT) for 24 h. The resultant dispersions were centrifuged at 1000 rpm for 30 min, and the supernatant were collected by pipette and then sprayed onto the inner wall of a heated rotating silica tube to synthesize aligned, compact films, and vertical heterostructures.

**Structure and property characterizations**. The microstructure of the synthesized films was characterized by SEM (Helios 600i) and XRD (Empyrean X'PERT). AFM (Bruker Dimension Fastscan) was used to identify the thickness of the 50-nm-thick films. X-ray photoelectron spectroscopy (XPS, Thermo Scientific ESCALAB 250Xi) and Fourier transform infrared spectroscopy (FTIR, Bruker Optics Tensor 37) were used to identify the content of iodine and SDS in the hybrid films, respectively. The mechanical and electrical properties of the films (at least 5 samples) were tested by DMA (TA-Q800) and Keithley 2500, respectively. The density of rGO and rGO/ SWCNT hybrid films was calculated based on their volume and mass, and their thickness and mass were measured by SEM and micro-balance, respectively.

**Fabrication and characterization of tape SCs**. The gel electrolyte of PVA/H₂SO₄ was prepared by mixing PVA power, water and concentrated sulfuric acid (1 g PVA/0.8 g H₂SO₄/10 g H₂O). First, two identical continuous rGO/SWCNT ribbons with thickness of ~39 μm were adhered in parallel onto a paper tape with a spacing of ~10 μm. After that, the polymeric gel electrolyte was dropped on top of the electrodes and then sealed with PDMS to obtain all-solid-state tape SCs. The electrochemical performance of the tape SCs was tested by CV and galvanostatic charge/discharge in a two-electrode configuration using the electrochemical workstation (CH1140C). The electrode capacitance of the SCs ($C_{electrode}$) was

calculated from the CV curves at different scan rates using

$$C_{electrode} = \frac{Q}{2V} = \frac{1}{2V\nu} \int_{V_-}^{V_+} i(V)dV, \qquad (1)$$

where $Q$ is the total voltammertic charge obtained by integration of the positive and negative sweeps ($i(V)$ is the current) in the CV curve, $\nu$ is the scan rate, and $V$ ($V = V_+ - V_-$) represents the scanned potential window of 1.0 V. The device capacitance ($C_{cell}$) was calculated by the galvanostatic charge/discharge curves at different current densities using

$$C_{cell} = i / \left(\frac{dV}{dt}\right), \qquad (2)$$

where $i$ is the discharging current and $dV/dt$ is the slope of the discharge curve.

The electrode and device volumetric capacitances of the SCs were obtained by the equations:

$$C_{electrode,V} = 2C_{cell}/V_{single}, \qquad (3)$$

$$C_{cell,V} = C_{cell}/V_{cell}, \qquad (4)$$

where $V_{single}$ and $V_{cell}$ are the volume of rGO/SWCNT single electrode and whole-all-solid-supercapacitor device, respectively. $V_{cell}$ includes the volumes of two electrodes and solid electrolyte. Since our electrode has rectangular cross-section, much less electrolyte (take up roughly ~25% of the whole-device volume) was needed as compared to the case of fibrous electrode (take up roughly ~40% of the whole-device volume as reported previously[41]), which is beneficial for improving the device volumetric performance.

The volumetric energy density of whole device ($E_{cell,V}$) was obtained by the equation

$$E_{cell,V} = C_{cell,V}\Delta E^2/(2 \times 3,600), \qquad (5)$$

where $\triangle E$ is the operating voltage window in volts.

The volumetric power density of the whole device ($P_{cell,V}$) was calculated from the galvanostatic curves at different charge/discharge current densities using the equation

$$P_{cell,V} = E_{cell,V} \times 3,600/t_{discharge}, \qquad (6)$$

where $t_{discharge}$ is the discharge time.

**Finite element simulation on the shear rate distribution.** A 2D finite turbulent flow model was built using COMSOL Multiphysics 5.1 to analyze the shear rate distribution of the rotating liquid. The size of the periodic geometric unit was ~$200 \times 200~\mu m^2$. In this model, a sliding wall boundary (liquid–solid boundary, non-slippery boundary) was set on the right wall of the tube, corresponding to the rotating wall, while a slip boundary was set on the left, corresponding to the liquid-air interface. The viscosity of the liquid was set on 10 Pa s. Three rotating rates, 500, 1000, and 1500 r min⁻¹, (corresponding to boundary velocity of 2.85 m s⁻¹, 5.7 m s⁻¹, and 8.55 m s⁻¹, respectively) were used to investigate the shear rate distribution in the rotating liquid.

**Data availability**. The data that support the findings of this study are available from the corresponding author upon request.

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

## Acknowledgements

This work was supported by the National Key R&D Program of China (No. 2016YFA0200101), National Science Foundation of China (Nos. 51325205, 51290273, 51521091, and U1633201), and Chinese Academy of Sciences (Nos. KGZD-EW-303-1, KGZD-EW-T06, 174321KYSB20160011, and XDPB06).

## Author contributions

W.R. conceived and supervised the project; J.Z. and W.R. designed the experiments; J.Z., W.S., Q.W. and X.Q. performed the experiments, J.Z. and W.R. analyzed the data; J.Z., H.C. and W.R. wrote the manuscript. All the authors discussed the results and commented on the manuscript.

## Additional information

**Competing interests:** The authors declare no competing interests.

