## [Peer Review File · Nature Communications]

Reviewers' comments:

Reviewer #1 (Remarks to the Author):

The authors report that the continuous centrifugal casting ("CCC") method is a very efficient method to fabricate highly compact and well aligned films composed of stacked and overlapped nanosheets such as graphene oxide (G-O; dispersed graphite oxide—graphite oxide is typically referred to as "GO" in the literature, the reviewer suggests it is best to use "G-O" for graphene oxide), and several other types of nanosheets. (Thus, the reviewer suggests it is also better to use rG-O for "reduced graphene oxide".)

The new approach for making films of stacked and overlapped nanosheets is very impressive.

One can note that the stress-strain behavior and large-area film formation seem similar to reference 21 (*Adv. Mater.* 27, 6708 (2015)). How would the authors more obviously contrast their films with those reported in reference 21 (or would they)?

The manuscript lacks a clear explanation for the mechanism of how the G-O film is made at the "sheet-by-sheet" level of assembly. The centrifugal force is related to the diameter of the cylinder, and the shear force is related to its rotational speed. The authors don't clearly explain the forces related to the alignment and overlapping of the G-O sheets.

1. A description is provided that says that according to changes in the RPM (revolutions per minute) the liquid flow changed, being either turbulent or laminar depending on the RPM values. A more detailed explanation of the fluid dynamics is needed.

2. In the part of the text where references 21 and 22 are cited there is a discussion of improvement of the G-O stacking by the removal of organic solvents (organosulfates, water) as studied by the FWHM value of a peak in XRD data. The authors note that as a function of rpm in their work reported here, the (002) XRD peak shifted from 10.36 to 11.46 degrees. How does this XRD peak shift represent the material characteristics of the films? Well-aligned films are shown in Figures 1d and 2c. The authors don't explain why these films are, as they say, more compact and with a higher degree of alignment of the stacked and overlapped G-O layers: but why?

3. Figure 2f shows the electrical conductivity of the G-O films after their treatment with HI (which deoxygenates them, leading to a large increase in electrical conductivity), thus rG-O films, according to the rpm value used for production of the precursor G-O film. The results are significantly better than the values obtained from the values from the literature references that are shown. Generally, electrical properties depend on 'material connectivity'. The rG-O film consists of physical inter-grain connections of the micron-scale rG-O sheets. What is the underlying physical mechanism(s) for a higher level of electrical conductivity?

Turning now to the production of films that are alternate stacking of carbon nanotubes and graphene oxide sheets (G-O) and then their chemical reduction to test their use as a supercapacitor, there are some serious issues that the authors need to address.

First though, I do note that this would appear to be a very promising way, indeed, to make a wide variety of films, very quickly, that would hold promise for a wide variety of applications.

However, that (alone) clearly does not warrant publication in *Nature Communications*, where there needs to also be a very clear advance in science (not engineering or a method for more rapid manufacture of films). The part of the manuscript describing the supercapacitor has some missing information and also it was necessary to analyze some of the presented data (but the authors should have provided such analyses) to assess the "quality" of the supercapacitor.

For example, according the galvanostatic charge-discharge curves, the fabricated devices have a Coulombic efficiency of about 70% (not very "good"). Also, the authors did not provide the electrode sizes, thicknesses, and masses. It is critical to know the thickness. 0.5 microns or 100

microns or 200 microns or? It is well known that a small loading of the active material (a small thickness--a small mass) results in high capacitance values, when a more realistic (that is, similar to commercial devices) loading can have much lower values. The authors need to provide "all" of the details for assessing this supercapacitor.

Another issue is removal of surfactant. SDS surfactant was used for the SWCNT dispersion. If one works through (but again, the authors should clearly state these things so that it is not up to the reader to try to gauge the amount...) the apparent quantity of SDS used, there should be a very large component of SDS in the rG-O/CNT (/surfactant?) film. All of this is really removed through the washing used? What is the experimental evidence for removal of the SDS?

I would readily recommend publication in a journal such as *Advanced Materials* even without the description of the fabrication and testing of the supercapacitor. The method of fabricating thin (but also thicker than readily obtained with other methods--which is quite important!) films of stacked and overlapped nanosheets (nanoplatelets, etc. : there is not an agreed upon terminology) is very exciting and will undoubtedly be very useful for much R&D in the future. But, for *Nature Communications*, there needs to be a deeper explanation of scientific issues.

Reviewer #2 (Remarks to the Author):

The article describes centrifugal casting of highly aligned and compact GO and rGO films with outstanding mechanical and electrochemical properties. Good integration of flakes leads to strong bonding, high electronic conductivity and capacitance values. The approach is novel and may be of practical importance. The mechanical properties are very impressive. The paper can be considered for publication after addressing the issues outlined below.

There are too many generic references in the abstract, which can be easily replaced by references to a couple of review articles. The same is the case in lines 42-46.

At the same time, it's difficult to understand the work because many details about materials used, properties measured, etc. are either missing or hidden in SI. Information about flake size, shear rate, electrolyte used and electrode thickness in capacitance measurements. At least basic details of manufacturing and technics should be provided in the main text.

What kind of bonding forms between layers? Just usual weak bonding? Why does the material then show better properties in that case?

What is the mechanism responsible for the close to 400 F/cm³ or F/g capacitance (Table 1)? Double-layer capacitance probably cannot alone explain those high values. Also, non-linear discharge in Fig. 3e supports presence of pseudocapacitance. The CVs look resistive and the energy efficiency not very high. I'm not sure if this is a result of unusual testing configuration (2 parallel stripes separated by 2 mm) or a material issue. The authors should address this problem. Conventional interdigitated electrodes would probably provide a better result.

The authors compare their material to oxides and other graphene based materials. However, they should also mention MXenes, which have higher electrical conductivity and higher capacitance, showing similar flexibility.

It is not clear to the review why increasing viscosity of the GO dispersion is supposed to lead to increased alignment and compaction of the films (lines 117-119).

The authors mention 1-micron GO sheets in Fig. 1 and then 20-micron sheets in line 137. Be specific about particle size used to produce films – it strongly affects the properties.

Provide units for shear rate and rpm information in Fig. 1b. The authors should also specify shear rate in line 85 and provide viscosity data for the shear rate used.

Insets in Fig. 3j are very small and therefore useless.

There are typos and the manuscript should be carefully edited:

Replace "langmuir-blodgett" with Langmuir-Blodgett,

An origami crane (line 131)

etc.

Reviewer #3 (Remarks to the Author):

The authors reported the use of centrifugal casting as a scalable method to produce highly aligned and compact 2D graphene oxide (GO) or reduced-GO (rGO) films. To the best of my knowledge, I didn't find any other literature using centrifugal casting to align GO nanosheets, so the method is relatively novel. The study also claimed that the thus formed thin films outperformed comparable counterparts with a variety of improved properties, such as better alignment and compaction than other reported GO films, higher strength of and conductivity than literature-reported rGO films, and higher energy density with the rGO/carbon nanotube hybrid film than lithium batteries. The presentation of such a wide a range of applications on one hand indicates the potential broad impact of such a synthesis method, but on the other hand compromised and sacrificed the opportunity of having in-depth or theoretical discussion of each claimed properties. It is suggested that the authors add more comprehensive discussion to explain why the centrifugal alignment results in the improved performance in different applications.

The rotating rate is apparently a very important parameter to a successfully aligned film. It effects shear stress and centrifugal force. An increased rotating rate could make the film more compact and aligned. But a higher rotating rate could lead to a turbulence flow which damage the alignment of GO nanosheets. So it's recommended that the authors provide an empirical equation that determines the maximum rotating rate considering the factors including solution viscosity, RHT diameter.

The authors demonstrated a smaller interlayer spacing and FWHM of GO obtained at 1500 r/min, as compared to those of the reported GO films. Though the higher rotating rate could improve the compact of GO films (Fig 1e), the authors should acknowledge the possible contribution of heating (80C in this work) to the smaller interlayer spacing. Heating could remove the residue water often seen in other methods like vacuum filtration and solvent evaporation, and even reduce GO films to some extent.

The authors should show the demonstration of nanoscale-thick film, which is appealing and has potential applications in a wide range of applications. The authors mentioned this method is applicable to make film with thickness from a few nanometers to hundreds of micrometers, but only demonstrated the formation and various applications of microscale-thick GO/rGO films, and the claim on forming nanometer thick film is made without experimental proof. As a matter of fact, it is often more challenging and desired to prepare a uniform nanoscale-thick film.

Line 130, "flexibility" should be changed to "flexible"

Response to reviewers' comments

Reviewer #1 (Remarks to the Author):

The authors report that the continuous centrifugal casting (“CCC”) method is a very efficient method to fabricate highly compact and well aligned films composed of stacked and overlapped nanosheets such as graphene oxide (G-O; dispersed graphite oxide—graphite oxide is typically referred to as “GO” in the literature, the reviewer suggests it is best to use “G-O” for graphene oxide), and several other types of nanosheets. (Thus, the reviewer suggests it is also better to use rG-O for “reduced graphene oxide”.)

Reply: We thank the reviewer very much for kind suggestion. We have used G-O and rG-O to replace GO and rGO, respectively, in the revised manuscript.

The new approach for making films of stacked and overlapped nanosheets is very impressive.

Reply: We thank the reviewer very much for positive comment.

One can note that the stress-strain behavior and large-area film formation seem similar to reference 21 (Adv. Mater. 27, 6708 (2015)). How would the authors more obviously contrast their films with those reported in reference 21 (or would they)?

Reply: We thank the reviewer very much for kind comment.

In Reference 21, the authors used (1) G-O nanosheets with low defect density as raw material, (2) then thermal annealing at 70 °C for 36 hours to form G-O hydrogel, and (3) finally cast drying the resultant hydrogel to produce G-O films, which typically takes 45 h to obtain a 9 – 11 μm thick film. HI solution was then used to reduce the G-O films to rG-O films. They found that the G-O nanosheets with low defect density is the prerequisite for thermal-driven G-O gelation and applying the gel-film-transformation approach to prepare strong G-O and rG-O films (614 MPa), and the low defect structure of G-O plays a critical role in the strength of the films. As shown in Reference 21, annealing conventional G-O dispersion under identical

conditions cannot induce gelation, and the tensile strength of the G-O films made by filtration or evaporation is lower than 120 MPa.

In sharp contrast, as shown in our manuscript, continuous centrifugal casting is a universal, scalable and highly efficient method to produce highly aligned and compact films of graphene and various other 2D materials. First, there are no strict requirements on the raw materials. Many kinds of 2D materials can be used to produce highly aligned and compact films even composites and heterostructures by this method. Second, it is highly efficient and easy to scale up. A 10- μm -thick G-O film ($\sim 30 \times 10 \text{ cm}^2$) can be produced within 1 min. However, it takes $\sim 81 \text{ h}$ to produce a 9 – 11 μm -thick G-O film with the similar size by the method reported in Reference 21. In order to further demonstrate the scalability and controllability of our method, we have synthesized $\sim 100 \text{ cm} \times 20 \text{ cm}$ sized thick and thin G-O films by using a rotating tube with a diameter of 33 cm (Fig. R1). Third, the films have excellent properties. For instance, the rG-O films synthesized by our method with conventional G-O nanosheets at 1,500 rpm show a tensile strength of 592 MPa, which is about twice the best value reported for the films made by conventional G-O nanosheets and comparable to those made by G-O nanosheets with low defect density reported in Reference 21 (614 MPa). More importantly, the properties of our films can be further improved by simply increasing the rotating rate. For instance, the rG-O films obtained at 2,500 r/min show a tensile strength of 662 MPa (Fig. R2), which is higher than those reported in Reference 21 (614 MPa).

In the revised manuscript, we have made the above points more clearly and added the above new data to further show the advantages of our method over that reported in Reference 21.

Figure R1. A meter-scale (a) ~ 10 μm -thick free-standing G-O film and (b) thin G-O film on PET produced with a rotating tube of 33 cm in inner diameter. (c) A thin G-O film on a 4 inch silicon wafer synthesized with the same conditions as those for the production of the film in **b**. (d) AFM image of the film in **c** and (e) the height profiles along the three lines in **d**, showing the thickness of ~ 50 nm.

Figure R2. (a) XRD patterns, (b) peak position and FWHM, and (c) stress-strain curves of the rG-O films produced with large G-O nanosheets (~ 20 μm) at different rotating rate.

The manuscript lacks a clear explanation for the mechanism of how the G-O film is made at the “sheet-by-sheet” level of assembly. The centrifugal force is related to the diameter of the cylinder, and the shear force is related to its rotational speed. The authors don’t clearly explain the forces related to the alignment and overlapping of the G-O sheets.

1. A description is provided that says that according to changes in the RPM (revolutions per minute) the liquid flow changed, being either turbulent or laminar depending on the RPM values. A more detailed explanation of the fluid dynamics is needed.

Reply: We thank the reviewer very much for kind suggestion.

In a liquid flow, shear stress is generated when there is velocity gradient in the direction that is perpendicular to the velocity direction. In our method, when the hollow tube is rotated, the first liquid layer that is adjacent to the tube inner surface will stick on the tube according to the non-slippery boundary condition, and thus has the same linear velocity with the inner surface of the rotating tube. The second liquid layer is then pulled by the first liquid layer to move along because of the viscosity of the liquid. Such disturbance generated by the tube rotation thus propagates from the liquid-solid interface (tube inner surface) deep into the liquid flow. The transient shear rate distribution at the beginning of the tube rotation (at 1 microsecond) was investigated by the software of CONSOL. It can be seen that the disturbance transports from the liquid-solid boundary into liquid immediately once the tube starts to rotate, generating uniform shear rate around the tube inner surface (Fig. 1b).

Based on the fluid mechanics, there are mainly two forces that control the behavior of the flow in the steady state, inertial force and viscose force [An introduction to fluid dynamics. Cambridge University Press, 1967]. The former can be considered as the dynamic sensitivity of flow, while the latter as the dissipation capability for the disturbance. Typically, the relative contribution of these two forces can be estimated by the Raynolds number, which is the ratio of inertial force to viscose force. In our

study, the liquid layer around the inner surface of a rotating hollow tube, typically considered as rimming flow, is a non-trivial example of a steady, two-dimensional, viscous flow with a free surface [*J. Fluid. Mech.* 190, 321-342 (1988); *J. Fluid. Mech.* 84, 145-165 (1978)]. The Reynolds number is expressed as:

$$Re \equiv \omega D^2 / \nu$$

where ω , D and ν are the angular velocity of the rotating tube (the corresponding rotating rate $r = 30\omega/\pi$ (r/min)), and the average thickness and kinematic viscosity of the flow, respectively.

When the Reynolds number is small, any disturbance, including the change of the tube rotation rate and any uncertainty of the flow boundary condition (such as the roughness of the inner hollow tube), can be dissipated and the flow exhibits characteristic of laminar flow. When the Reynolds number is large, any disturbance will be amplified and eventually results in the generation of eddies, and the flow exhibits characteristic of turbulence. In fact, the turbulence flow is extensively considered as the mechanism for dispersion and mixing. Therefore, the flow should be controlled within the laminar flow in order to align the 2D nanosheets.

As discussed above, in order to guarantee the liquid flow is located in the region of laminar flow, Reynolds number must be depressed, that is, viscose effects dominate the behavior of the flow. The critical Reynolds number for the transition between laminar flow and turbulent flow is dependent on the physical properties of the flow. Here, we take the critical Reynolds number to be one, to roughly estimate the domains corresponding to laminar flow phase and turbulent phase, and the phase diagram obtained is shown in Fig. R3. Note that the viscosity of G-O dispersion is increased along with the evaporation of water induced by heating during the continuous centrifugal casting process. As shown in Fig. R3, the G-O flow is always limited in the laminar flow region for all the rotating rates used in our experiments (500, 1000, 1500, and 2500 r/min) even the viscosity increase is not considered. Moreover, it is still limited in the laminar flow region even though the viscosity keeps unchanged and

the angular velocity reaches 1,000 rad/s, corresponding to the tube rotating rate of $\sim 9,500$ r/min.

Figure R3. The phase diagram for the laminar and turbulent phase. The bottom-right and up-left corresponds to the turbulent flow and laminar flow, respectively. The blue dots represent the experimental conditions that are used in our study. In fact, these blue dots should move further away from the phase transition interface since the real viscosity should be much higher than the original value because of the water evaporation during the continuous centrifugal casting process.

We then analyzed the forces that are applied to 2D nanosheets in the continuous centrifugal casting process. As discussed above, a liquid layer around the inner surface of a rotating hollow tube, typically considered as rimming flow, is a non-trivial example of a steady, two-dimensional, viscous flow with a free surface. As shown in Fig. R4, we considered a horizontally placed circular hollow cylinder with inside radius R_0 , which rotates at a constant angular velocity (ω) around its axis. Polar coordinates were used: Θ and \mathfrak{R} are the angular azimuthal coordinate and radial coordinate, respectively, and e_θ and e_r are the corresponding unit tangent vectors. The cylinder is partially filled with Newtonian liquid which is distributed in a layer around the inner wall. Strictly speaking, the thickness of the liquid layer varies with the cylinder and is a function of θ , that is $H(\theta)$, because of the influence of the local gravitational acceleration (which is denoted as G along the vertical direction of j).

Since the gas occupying the rest of the volume in the cylinder is inviscid, the viscous stress exerted by the relatively rarefied gas at the gas/liquid interface is negligible, and thus can be modeled as by a uniform pressure P_a .

Figure R4. Schematic of a rotating hollow cylinder partially filled with a liquid layer.

Based on the above analysis, this trimming flow can be modeled mathematically as follows:

$$\left\{ \begin{array}{l} \mathbf{V} \cdot \nabla \mathbf{V} = -\rho^{-1} \nabla P + \nu \nabla^2 \mathbf{V} - G\mathbf{j} \\ \nabla \cdot \mathbf{V} = 0. \end{array} \right\}$$

where $V(R)$ and $P(R)$ are liquid velocity and pressure fields, respectively. The density ρ , viscosity μ , kinematic viscosity $\nu = \mu/\rho$ and surface tension σ of the liquid are all assumed uniform.

The no-slip boundary condition at the liquid-wall interface is:

$$\mathbf{V} = \Omega R_0 \mathbf{e}_\theta \quad \text{at} \quad \mathbf{R} = R_0 \mathbf{e}_r.$$

The balance of traction at the liquid/gas interface gives the boundary condition:

$$(P - P_a)\mathbf{n} - \mu \left[\nabla \mathbf{V} + (\nabla \mathbf{V})^T \right] \cdot \mathbf{n} + 2\sigma K \mathbf{n} = 0 \quad \text{at} \quad \mathbf{R} = -(R_0 - H)\mathbf{e}_r.$$

where \mathbf{n} is the unit normal to the interface, directed to the gas, and K is the mean curvature of the interface:

$$K = \frac{1 + 2[H_\theta / (R_0 - H)]^2 + H_{\theta\theta} / (R_0 - H)}{2(R_0 - H)\{1 + [H_\theta / (R_0 - H)]^2\}^{\frac{3}{2}}}$$

Since no liquid can cross the interface, the velocity along the normal direction should be zero:

$$\mathbf{V} = \omega R_0 \mathbf{e}_\theta \quad \text{at} \quad \mathbf{R} = R_0 \mathbf{e}_r$$

It is obvious that the solution for the above equations must be periodic with respect to the coordinate of θ with a period of 2π .

The average thickness of the liquid layer (D) is given by

$$2R_0D - D^2 = \frac{1}{2\pi} \int_0^{2\pi} (2R_0H - H^2) dX$$

For the high rotation rate, G can be neglected, and thus we assume $G = 0$.

An analytical solution can be obtained:

$$\left. \begin{aligned} \mathbf{V} &= \omega(R_0 - Y) \mathbf{e}_\theta \\ P - P_a &= \rho \Omega^2 \left[R_0(D - Y) - \frac{1}{2}(D^2 - Y^2) \right] - \frac{\sigma}{R_0 - D} \\ H &= D. \end{aligned} \right\}$$

The derived shear stress for this Newtonian flow is:

$$\boldsymbol{\sigma}_{\text{shear}} = \mu \omega \mathbf{e}_\theta$$

During the continuous centrifugal casting process, the 2D nanosheets are also subjected to centrifugal force at the same time. The centrifugal stress that is applied on 2D nanosheets is given by:

$$\boldsymbol{\sigma}_{\text{centrifugal}} = \rho_g t_g R_0 \omega^2 \mathbf{e}_r$$

where ρ_g and t_g are the density and thickness of 2D nanosheets, respectively.

The shear stress is along the tangential direction of the rotating tube, which not only can align but also can smooth the 2D nanosheets. As reported previously, a shear stress of ~ 20 Pa is sufficient to align the G-O nanosheets [PNAS. 113, 11088-11093

(2016)]. We estimated the shear stress applied to G-O nanosheets in the G-O dispersion used, which has a viscosity of 1.26 Pa.s, at a low rotating rate of 500 r/min. The shear stress obtained is ~ 66 Pa. During our CCC process, the real shear stress should be much larger since the viscosity of the liquid is increased with the evaporation of water induced by heating. The centrifugal stress is along the radial direction of the rotating tube, which provide the force for condense stacking of parallel aligned smooth G-O nanosheets. In addition, the evaporation of water induced by low-temperature heating can help the condensation. Therefore, highly aligned and compact G-O films are formed by continuous centrifugal casting.

Based on the above equations, the shear stress increases linearly with the rotating rate and viscosity of the liquid, and the centrifugal stress increases with the rotating rate and the diameter of rotating tube. For instance, the shear stress is increased from 66 to 131 and 197 Pa, respectively, when the rotating rate is increased from 500 to 1,000 and 1,500 r/min. Because of the flow is always in the region of laminar flow even when the tube rotating rate reaches $\sim 9,500$ r/min, the alignment and compaction of G-O films are improved by increasing the rotating rate as shown in our original manuscript. To further demonstrate the influence of shear stress and centrifugal stress, we further increased the shear stress (329 Pa) and centrifugal stress by increasing the rotating rate to 2,500 r/min, and consequently improved the alignment, compaction and mechanical strength of G-O and rG-O films, as shown in Fig. R2 and R5.

Figure R5. (a) XRD patterns, (b) peak position and FWHM, and (c) stress-strain curves of the G-O films produced with small G-O nanosheets (~ 1 μm) at different rotating rate.

2. In the part of the text where references 21 and 22 are cited there is a discussion of improvement of the G-O stacking by the removal of organic solvents (organosulfates, water) as studied by the FWHM value of a peak in XRD data. The authors note that as a function of rpm in their work reported here, the (002) XRD peak shifted from 10.36 to 11.46 degrees. How does this XRD peak shift represent the material characteristics of the films? Well-aligned films are shown in Figures 1d and 2c. The authors don't explain why these films are, as they say, more compact and with a higher degree of alignment of the stacked and overlapped G-O layers: but why?

Reply: We thank the reviewer very much for kind comments.

Recently, XRD has been widely used to characterize the structure of assembled G-O-based films. Based on the bragg's law, $2d\sin\theta = \lambda$, the position of XRD peak (θ) determines the interlayer distance (d), and the full width at the half maximum (FWHM) reflects the ordering, which is correlated with the alignment and flatness of the nanosheets [e.g., *Nature* 448, 457-460, (2007); *Adv. Mater.* 20, 3557-3561, (2008); *ACS Nano* 6, 10708-10719, (2012); *Nat. Commun.* 7, 10891, (2016); *Nat. Mater.* 16, 1198-1202, (2017)]. As shown in our original manuscript, the XRD peak shifts from 10.36 to 11.46 degree and the FWHM dramatically decreases from 1.72° to 0.53° when the rotating rate is increased from 0 to 1,500 r/min. These XRD peak shifts suggest that the interlayer distance of G-O nanosheets decrease along with a better ordering. That is, the G-O sheets in the films become more compact and aligned.

As shown in our previous reply, during the continuous centrifugal casting process, the G-O nanosheets are subjected to shear stress and centrifugal stress at the same time. The shear stress is along the tangential direction of the rotating tube, which not only can align but also can smooth the G-O nanosheets. The centrifugal stress is along the radial direction of the rotating tube, which provide the force for condense stacking of parallel aligned smooth G-O nanosheets. In addition, the evaporation of water induced by low-temperature heating can help the condensation. As shown in the equations of shear stress and centrifugal stress, the shear stress increases linearly with the rotating rate and viscosity of the liquid and the centrifugal stress increases with the

rotating rate and the diameter of rotating tube. Because of the flow is always in the region of laminar flow even when the tube rotating rate reaches ~9,500 r/min, the alignment and compaction of G-O films are greatly improved by increasing the rotating rate from 500 to 1,500 r/min, as shown in our original manuscript. When the rotating rate is increased to 2,500 r/min, the alignment, compaction and mechanical strength of G-O and rG-O films are further improved (Fig. R2 and R5), which confirms the role of shear stress and centrifugal stress.

3. Figure 2f shows the electrical conductivity of the G-O films after their treatment with HI (which deoxygenates them, leading to a large increase in electrical conductivity), thus rG-O films, according to the rpm value used for production of the precursor G-O film. The results are significantly better than the values obtained from the values from the literature references that are shown. Generally, electrical properties depend on ‘material connectivity’. The rG-O film consists of physical inter-grain connections of the micron-scale rG-O sheets. What is the underlying physical mechanism(s) for a higher level of electrical conductivity?

Reply: We thank the reviewer very much for kind comment.

As the reviewer mentioned, the electrical conductivity of rG-O film strongly depends on the connection between the stacked adjacent nanosheets, which determines the contact resistance. Considering the continuous centrifugal casting process, the adjacent rG-O nanosheets in the films obtained are attracted each other mainly by Van der Waal’s force. The van der Waals force can be estimated by Lifshitz’s formula $F = S \frac{A_{Ham}}{6\pi d^3}$ [*Chem. Mater.* 19, 4396–4404 (2007)], where A_{Ham} is the Hamaker coefficient, d and S are the interlayer distance and contact area of the adjacent rG-O nanosheets, respectively. It can be seen from this formula that the binding force is highly sensitive to the interlayer distance and proportional to the contact area between adjacent reduced G-O layers.

As mentioned above, for G-O based films, the position of XRD peak (θ) determines the interlayer distance (d), while the FWHM reflects the ordering, which is correlated

with the alignment and flatness of nanosheets. Normally, the better alignment can increase the contact and interaction [*Nature* 448, 457-460, (2007)]. We have performed XRD measurements on the rG-O films prepared at different rotation rate with same G-O nanosheets as raw material. Similar to the G-O films, the XRD peak upshifts together with a great decrease in FWHM with increasing the rotation rate (Fig. R2a, b). This indicates that the connection and contact between stacked adjacent rG-O nanosheets are improved. As a consequence, the electrical conductivity of the films increases with increasing the rotating rate.

As shown in Table R1, it is worth noting that the highly conductive rG-O films prepared with 20- μm -sized G-O nanosheets at rotating rate of 1,500 and 2,500 r/min show smaller interlayer distance and better alignment than those listed for comparison in Fig. 2h. Therefore, we suggest that the better connection and contact in our films are the main reasons for the higher level of electrical conductivity.

We have added the above discussions and Table R1 in the revised manuscript.

Table R1. Structure, strength and electrical conductivity comparison of the rG-O films synthesized by different methods.

Synthesis method	lateral size (μm)	XRD Peak (degree)	FWHM (degree)	Strength (MPa)	Conductivity (S cm^{-1})	Reference
VF	5	NA	NA	20	190	ACS Nano 6, 10708 (2012)
VF	33	NA	NA	24	380	
VF	60	NA	NA	26	480	
VF	93	NA	NA	28	500	
SA	5	24.4	1.31	162	298	Carbon 48, 4466 (2010)
VF	1.3	23.37	4.97	150	40	Adv. Mater. 20, 3557 (2008)
VF@220 °C	1.3	25.66	1.7	293	100	
VF@500 °C	1.3	26.15	1.55	90	295	
VF	2.5	NA	NA	117	45	ACS Nano 8, 9511 (2014)
VF	112	24.12	3.5	66	200	Adv. Mater. 26, 4521 (2014)
SE	22	24.62	3.56	170	200	Chem. Mater. 26, 6786 (2014)

VF	NA	NA	NA	118	50	Angew. Chem. 25, 3750 (2013)
CCC@1500	20	25.7	1.13	591	602	This work
CCC@2500	20	25.94	1.02	662	647	This work

Turning now to the production of films that are alternate stacking of carbon nanotubes and G-O sheets (G-O) and then their chemical reduction to test their use as a supercapacitor, there are some serious issues that the authors need to address.

First though, I do note that this would appear to be a very promising way, indeed, to make a wide variety of films, very quickly, that would hold promise for a wide variety of applications.

However, that (alone) clearly does not warrant publication in Nature Communications, where there needs to also be a very clear advance in science (not engineering or a method for more rapid manufacture of films). The part of the manuscript describing the supercapacitor has some missing information and also it was necessary to analyze some of the presented data (but the authors should have provided such analyses) to assess the “quality” of the supercapacitor.

For example, according the galvanostatic charge-discharge curves, the fabricated devices have a Coulombic efficiency of about 70% (not very “good”). Also, the authors did not provide the electrode sizes, thicknesses, and masses. It is critical to know the thickness. 0.5 microns or 100 microns or 200 microns or? It is well known that a small loading of the active material (a small thickness--a small mass) results in high capacitance values, when a more realistic (that is, similar to commercial devices) loading can have much lower values. The authors need to provide “all” of the details for assessing this supercapacitor.

Reply: We thank the reviewer very much for kind comments and suggestions.

In order to clarify the origin of the low Coulombic efficiency, we have investigated the amount of SDS and iodine in the rG-O/SWCNT hybrid films by using FTIR spectroscopy and XPS, respectively. The FTIR peaks at 2,853 and 2,992 cm^{-1} correspond to the vibrational modes of methyl and methylene group in SDS, respectively, which are used to identify qualitatively the amount of residual SDS. In our original manuscript, the hybrid films used for supercapacitors were washed for 0.5 h. It can be seen that they still contain some SDS (Fig. R6). After thorough washing over 18 h, the two FTIR peaks become almost invisible, indicating that most of the SDS are removed. XPS measurements show that the content of residual iodine decreases from 3.55% to 0.12% when extending the washing time from 0.5 to 18 h (Fig. R7). As shown in the Fig. R8, the films obtained after through washing exhibit a high Coulombic efficiency of ~98% and a slightly decreased volumetric capacitance (~1%). In the revised manuscript, we have replaced all the electrochemical performances with those based on the thoroughly washed hybrid films.

Indeed, the performance of electrode is sensitive to the thickness of electrode, when it is in the range of nanometer thickness. As the reviewer mentioned, a small loading of the active material (small thickness-- small mass) results in high capacitance values. In order to avoid such exaggeration, it was suggested by Prof. Ruoff that the thickness of the electrode is preferred to be more than 10 μm [*Energy Environ. Sci.* 3, 1294-1301 (2010)], which is comparable to that of the commercially available supercapacitors. The thickness of the electrodes used in our study is ~39 μm , and the width, length and mass are 2 mm, 50 mm and ~6.4 mg, respectively. In the revised manuscript, we have given all these information of the electrodes.

Figure R6. FTIR spectra of the rG-O/SWCNT hybrid film soaked in DI water for different time followed by ethanol washing for 2 h. The peaks at 2853 and 2992 cm⁻¹ correspond to the vibrational modes of methyl and methylene group in SDS, respectively, which are used here to measure qualitatively the amount of residual SDS. Both of the two peaks diminish with increasing the soaking time in water.

Figure R7. XPS spectra of the rG-O/SWCNT hybrid film soaked in DI water for different time followed by ethanol washing for 2 h.

Figure R8. Galvanostatic charge/discharge curves measured between 0 and 1 V at current density from 114 to 1,894 mA/cm³, and the electrode and device volumetric capacitance as a function of current density.

Another issue is removal of surfactant. SDS surfactant was used for the SWCNT dispersion. If one works through (but again, the authors should clearly state these things so that it is not up to the reader to try to gauge the amount...) the apparent quantity of SDS used, there should be a very large component of SDS in the rG-O/CNT (/surfactant?) film. All of this is really removed through the washing used? What is the experimental evidence for removal of the SDS?

Reply: We thank the reviewer very much for kind comments.

We investigated the amount of SDS in the rG-O/SWCNT hybrid films by using FTIR spectroscopy. The FTIR peaks at 2853 and 2992 cm⁻¹ correspond to the vibrational modes of methyl and methylene group in SDS, respectively, which are used to identify qualitatively the amount of residual SDS. In our original manuscript, the hybrid films used for supercapacitors were washed for 0.5 h. It can be seen that they still contain some SDS (Fig. R6). After thorough washing the films over 18 h, the two FTIR peaks become almost invisible, indicating that most of the SDS residues are

removed. We also investigated the residual iodine in the hybrid films using XPS. It was found that the content of residual iodine decreases from 3.55% to 0.12% when extending the washing time from 0.5 to 18 h (Fig. R7).

Based on these studies, we fabricated supercapacitors using thoroughly washed rG-O/SWCNT hybrid films and measured their performances. In the revised manuscript, we have added the FTIR and XPS spectra and replaced the electrochemical performances of supercapacitors with those based on thoroughly washed rG-O/SWCNT hybrid films.

I would readily recommend publication in a journal such as *Advanced Materials* even without the description of the fabrication and testing of the supercapacitor. The method of fabricating thin (but also thicker than readily obtained with other methods--which is quite important!) films of stacked and overlapped nanosheets (nanoplatelets, etc. : there is not an agreed upon terminology) is very exciting and will undoubtedly be very useful for much R&D in the future. But, for *Nature Communications*, there needs to be a deeper explanation of scientific issues.

Reply: We thank the reviewer very much for the high recognition on our method. According to the reviewer's valuable suggestions, as shown above, we have provided deeper explanation of the scientific issues related to the results presented in the manuscript, including the fluid dynamics analysis during continuous centrifugal casting process, the formation mechanism of highly aligned and compact films, the physical origin of the better performance of rG-O films, and more thorough analyses on the rG-O/SWCNT hybrid films and the performance of supercapacitors. In addition, we have performed more experiments to synthesize meter-scale tens of micrometers thick and nanometers thin G-O films and further improve the properties of the films, which further show the advantages of our method over the present methods in terms of scalability, controllability and performances. All the experimental details have been given in the revised manuscript as well. Once again, we would like to thank the reviewer for the constructive and valuable comments which have helped us to significantly improve the quality of the manuscript.

Reviewer #2 (Remarks to the Author):

The article describes centrifugal casting of highly aligned and compact GO and rGO films with outstanding mechanical and electrochemical properties. Good integration of flakes leads to strong bonding, high electronic conductivity and capacitance values. The approach is novel and may be of practical importance. The mechanical properties are very impressive. The paper can be considered for publication after addressing the issues outlined below.

Reply: We thank the reviewer very much for positive comments.

There are too many generic references in the abstract, which can be easily replaced by references to a couple of review articles. The same is the case in lines 42-46.

Reply: According to the reviewer's kind suggestion, we have added some related review articles in the abstract and introduction part.

At the same time, it's difficult to understand the work because many details about materials used, properties measured, etc. are either missing or hidden in SI. Information about flake size, shear rate, electrolyte used and electrode thickness in capacitance measurements. At least basic details of manufacturing and technics should be provided in the main text.

Reply: We thank the reviewer very much for constructive suggestions.

We used GO nanosheets with a mean lateral size of $\sim 1 \mu\text{m}$ and SWCNTs with an average diameter of $\sim 2 \text{ nm}$ as raw materials and a rotating rate of 1,500 rpm to fabricate highly aligned and compact hybrid films. In the supercapacitors, the electrolyte is PVA/H₂SO₄ gel and the electrodes have a thickness of $\sim 39 \mu\text{m}$, width of 2 mm, length of 50 mm and mass of $\sim 6.4 \text{ mg}$.

Besides the above details, we have checked the manuscript very carefully and given all the other experimental details with the basic details in the results and discussion parts. We have also moved the Method part from SI to the main text.

What kind of bonding forms between layers? Just usual weak bonding? Why does the material then show better properties in that case?

Reply: We thank the reviewer very much for kind comment.

During the continuous centrifugal casting process, the GO nanosheets are subjected to shear stress and centrifugal stress at the same time. The shear stress is along the tangential direction of the rotating tube, which not only can align but also can smooth the GO nanosheets. The centrifugal stress is along the radial direction of the rotating tube, which provide the force for condense stacking of parallel aligned smooth GO nanosheets. In addition, the evaporation of water induced by low-temperature heating can help the condensation. These three factors make the nanosheets well aligned and highly compact in the films. Therefore, we suggest that the adjacent layers are attracted each other mainly by Van der Waal's force in the GO and rGO films, which are similar to those synthesized by other methods.

The van der Waals force can be estimated by Lifshitz's formula $F = S \frac{A_{Ham}}{6\pi d^3}$, where A_{Ham} is the Hamaker coefficient, d and S are the interlayer distance and contact area of the adjacent rGO nanosheets. It can be seen from this formula that the binding force is highly sensitive to the interlayer distance and proportional to the contact area between adjacent rGO layers. For GO-based films, the position of XRD peak (θ) determines the interlayer distance (d), while the FWHM reflects the ordering, which is correlated to the alignment degree. Normally, the better alignment can increase the contact and interaction [*Nature* 448, 457-460, (2007)]. We have performed XRD measurements on the rGO films prepared at different rotation rate with same GO nanosheets as raw material. Similar to the GO films, the XRD peak upshifts together with a great decrease in FWHM with increasing the rotation rate (Fig. R2a, b). This indicates that the connection and van der Waals force between stacked adjacent rGO nanosheets are improved. As a consequence, both the mechanical strength and electrical conductivity of the films increases with increasing the rotating rate.

As shown in Table R1, it is worth noting that the rGO films prepared with 20- μm -sized GO nanosheets at rotating rate of 1,500 and 2,500 r/min show smaller interlayer distance and better alignment than those listed for comparison in Fig. 2h. Therefore, we suggest that the better alignment and higher compaction are the main reasons for the better properties than those reported in the literature.

We have added the above discussions and Table R1 in the revised manuscript.

Table R1. Structure, strength and electrical conductivity comparison of the rGO films synthesized by different methods.

Synthesis method	lateral size (μm)	XRD Peak (degree)	FWHM (degree)	Strength (MPa)	Conductivity (S cm^{-1})	Reference
VF	5	NA	NA	20	190	ACS Nano 6, 10708 (2012)
VF	33	NA	NA	24	380	
VF	60	NA	NA	26	480	
VF	93	NA	NA	28	500	
SA	5	24.4	1.31	162	298	Carbon 48, 4466 (2010)
VF	1.3	23.37	4.97	150	40	Adv. Mater. 20, 3557 (2008)
VF@220 °C	1.3	25.66	1.7	293	100	
VF@500 °C	1.3	26.15	1.55	90	295	
VF	2.5	NA	NA	117	45	ACS Nano 8, 9511 (2014)
VF	112	24.12	3.5	66	200	Adv. Mater. 26, 4521 (2014)
SE	22	24.62	3.56	170	200	Chem. Mater. 26, 6786 (2014)
VF	NA	NA	NA	118	50	Angew. Chem. 25, 3750 (2013)
CCC@1500	20	25.7	1.13	591	602	This work
CCC@2500	20	25.94	1.02	662	647	This work

What is the mechanism responsible for the close to 400 F/cm^3 or F/g capacitance (Table 1)? Double-layer capacitance probably cannot alone explain those high values. Also, non-linear discharge in Fig. 3e supports presence of pseudocapacitance. The CVs look resistive and the energy efficiency not very high. I'm not sure if this is a

result of unusual testing configuration (2 parallel stripes separated by 2 mm) or a material issue. The authors should address this problem. Conventional interdigitated electrodes would probably provide a better result.

Reply: We thank the reviewer for kind comment and suggestion.

In order to clarify the origin of the high capacitance, we measured the electrochemical performance of rGO/SWCNTs hybrid film in liquid electrolyte of 1 M H₂SO₄ using a three-electrode system. As shown in Fig. R9, the hybrid film shows non-rectangular CV curve with clear redox peaks, indicating the presence of pseudocapacitance. This is consistent with the non-linear discharge characteristic observed in Fig. 3e.

During the fabrication process of rGO/SWCNT hybrid films, we used SDS to disperse SWCNTs and HI to reduce GO nanosheets. We then investigated the amount of SDS and iodine in the rGO/SWCNT hybrid films by using FTIR and XPS, respectively. The FTIR peaks at 2,853 and 2,992 cm⁻¹ correspond to the vibrational modes of methyl and methylene group in SDS, respectively, which are used to identify qualitatively the amount of residual SDS. In our original manuscript, the hybrid films used for supercapacitors were washed for 0.5 h. It can be seen that they still contain some SDS (Fig. R6). After thorough washing over 18 h, the two FTIR peaks become almost invisible, indicating that most of the SDS are removed. XPS measurements show that the content of residual iodine decreases from 3.55% to 0.12% when extending the washing time from 0.5 to 18 h (Fig. R7).

As shown in Fig. R9, the thoroughly washed hybrid film measured in liquid electrolyte of 1 M H₂SO₄ in 3-electrode configuration show rectangular CV curves. We then fabricated all-solid-state tape supercapacitors using thoroughly washed hybrid films and gel electrolyte (PVA/H₂SO₄). As shown in Fig. R8, they show more linear discharge and a high Coulombic efficiency of ~98%. These results indicate that the low energy efficiency and pseudocapacitance originate from the residual SDS and iodine in the samples. Due to the elimination of pseudocapacitance contribution after

thorough washing, the gravimetric capacitance decreases from ~391 F/g to ~255 F/g. However, it is worth noting that the density of the hybrid film increases from 1.06 to 1.59 g/cm³ after thorough washing (Table R2), indicating that the removal of both SDS and iodine improves the compaction of the film. As a result, the thoroughly washed hybrid film still remains a very high volumetric capacitance of 407 F/cm³.

Figure 3c in the original manuscript is to demonstrate that the hybrid ribbons can be patterned in parallel on the paper tape. The real spacing of the two parallel ribbons (2 mm in width for each) used to construct tape supercapacitors is 10 μm.

In the revised manuscript, we have replaced Fig. 3c and the electrochemical performances of the supercapacitors with those based on thoroughly washed films.

Figure R9. Electrochemical performance of the rGO/SWCNT hybrid film in liquid electrolyte of 1 M H₂SO₄, tested in 3-electrode configuration.

Table R2. Properties of rGO/SWCNT hybrid films synthesized with different mass ratio of rGO to SWCNT.

Sample	GO : SWCNT mass ratio	Density (g cm ⁻³)	Tensile strength (MPa)	Conductivity (S cm ⁻¹)	Specific surface area (m ² g ⁻¹)	Volumetric capacitance (F cm ⁻³)	Gravimetric capacitance (F g ⁻¹)
rGO	---	2.05	296	467	< 1	163.1	79.4
rGO/SWCNT-4	4:1	1.79	121	231	< 1	210	117
rGO/SWCNT-2	2:1	1.67	94	203	46	267	159
rGO/SWCNT-1	1:1	1.59	71	184	129	407	255

The authors compare their material to oxides and other graphene based materials. However, they should also mention MXenes, which have higher electrical conductivity and higher capacitance, showing similar flexibility.

Reply: We thank the reviewer very much for kind suggestion. MXenes are a new class of 2D materials with many intriguing properties, and in particular, have been widely demonstrated to have a great potential for electrochemical energy storage. In the revised manuscript, we have added several representative papers about MXenes [*Adv. Mater.* 23, 4248-4253, (2011); *Nat. Rev. Mater.* 2, 16098, (2017); *Nature* 516, 78-81, (2014)] and the following discussion.

MXenes is a new class of 2D materials with many intriguing properties such as good dispersion in water, excellent electrical conductivity and high volumetric capacitance [*Adv. Mater.* 23, 4248-4253, (2011); *Nat. Rev. Mater.* 2, 16098, (2017)]. For instance, the free-standing $Ti_3C_2T_x$ paper electrodes show a volumetric capacitance exceeding 900 F cm^{-3} in 1 M H_2SO_4 [*Nature* 516, 78-81, (2014)]. It is reasonable to expect that highly aligned and compact MXenes papers can be synthesized by the CCC method to realize a super-high volumetric capacitance.

It is not clear to the review why increasing viscosity of the GO dispersion is supposed to lead to increased alignment and compaction of the films (lines 117-119).

Reply: We thank the reviewer very much for kind comment.

As shown in the reply to Reviewer 1's comments, we have deeply analyzed the fluid dynamics in the continuous centrifugal casting process. In order to synthesize highly aligned 2D nanosheets films, the liquid should satisfy the following two requirements at the same time.

First, the liquid should be limited in the region of laminar flow. Therefore, the Reynolds number must be depressed, which is expressed as:

$$Re \equiv \omega \rho D^2 / \mu$$

where ω , ρ , D and μ are the angular velocity of the rotating tube and the density, thickness and viscosity of the liquid, respectively.

Second, the shear stress should be as high as possible, which is expressed as:

$$\sigma_{\text{shear}} = \mu\omega e_{\theta}$$

where μ , ω , and e_{θ} are the viscosity of the liquid and the angular velocity and unit tangent vector of the rotating tube, respectively.

Based on the above two equations, it can be seen that increasing viscosity of the GO dispersion not only can decrease the Reynolds number but also can increase the shear stress, both of which are beneficial for increasing the alignment of the films. Moreover, the shear stress can smooth the GO nanosheets. During the continuous centrifugal casting process, the GO nanosheets are also subjected to centrifugal force at the same time, which is responsible for the compaction of GO. Compared to the crumpled structure, the well-aligned smooth 2D nanosheets are beneficial for condense stacking in parallel fashion. Therefore, increasing viscosity of the GO dispersion can lead to increased alignment and compaction of the films.

The authors mention 1-micron GO sheets in Fig. 1 and then 20-micron sheets in line 137. Be specific about particle size used to produce films – it strongly affects the properties.

Reply: We thank the reviewer very much for kind suggestion.

The GO films shown in Fig. 1 were synthesized by GO sheets with an average lateral size of 1 μm . As the reviewer mentioned, the lateral size of GO sheets has a strong influence on the properties of the assembled films. In order to synthesize conductive graphene films, we used two kinds of GO sheets with an average lateral size of 1 μm and 20 μm as raw materials. As shown in Fig. 2, the graphene films made by large GO sheets show much higher electrical conductivity and mechanical strength than those made by small sheets. We have specified the lateral size of all the GO sheets that were used to synthesize films in the revised manuscript.

Provide units for shear rate and rpm information in Fig. 1b. The authors should also specify shear rate in line 85 and provide viscosity data for the shear rate used.

Reply: We thank the reviewer very much for kind suggestion. In Fig. 1b, we used international unit for shear rate, which is $1/m$, and the rotating rate of the tube is 500 (left panel), 1,000 (middle panel) and 1,500 r/min (right panel). The viscosity of the liquid used in the simulation is 10 Pa·s. We have given all these information in the revised manuscript.

Insets in Fig. 3j are very small and therefore useless.

Reply: We have replaced the inset with larger and clearer figures.

There are typos and the manuscript should be carefully edited:

Replace “langmuir-blodgett” with Langmuir-Blodgett,

An origami crane (line 131)

etc.

Reply: According to the reviewer’s kind suggestion, we have checked the manuscript very carefully and corrected these mistakes and others.

Reviewer #3 (Remarks to the Author):

The authors reported the use of centrifugal casting as a scalable method to produce highly aligned and compact 2D graphene oxide (GO) or reduced-GO (rGO) films. To the best of my knowledge, I didn't find any other literature using centrifugal casting to align GO nanosheets, so the method is relatively novel. The study also claimed that the thus formed thin films outperformed comparable counterparts with a variety of improved properties, such as better alignment and compaction than other reported GO films, higher strength of and conductivity than literature-reported rGO films, and higher energy density with the rGO/carbon nanotube hybrid film than lithium batteries. The presentation of such a wide a range of applications on one hand

indicates the potential broad impact of such a synthesis method, but on the other hand compromised and sacrificed the opportunity of having in-depth or theoretical discussion of each claimed properties. It is suggested that the authors add more comprehensive discussion to explain why the centrifugal alignment results in the improved performance in different applications.

Reply: We thank the review very much for position comments and kind suggestions.

As shown in the reply to reviewer 1, we have deeply analyzed the fluid dynamics in the continuous centrifugal casting process. During continuous centrifugal casting process, both shear stress and centrifugal stress are subjected to the GO nanosheets. The shear stress is along the tangential direction of the rotating tube, which not only can align but also can smooth the GO nanosheets. The centrifugal stress is along the radial direction of the rotating tube, which provide the force for condense stacking of parallel aligned smooth GO nanosheets. In addition, the evaporation of water induced by low-temperature heating can help the condensation. Therefore, highly aligned and compact GO films are formed by continuous centrifugal casting. According to the equations of shear stress ($\sigma_{\text{shear}} = \mu \omega e_{\theta}$) and centrifugal stress ($\sigma_{\text{centrifugal}} = \rho_g t_g R_0 \omega^2 e_r$), the shear stress increases linearly with the rotating rate (ω) and viscosity (μ) of the liquid and the centrifugal stress increases with rotating rate (ω) and the diameter of rotating tube (R_0). Because of the flow is always in the region of laminar flow even when the tube rotating rate reaches $\sim 9,500$ r/min, the alignment and compaction of G-O films are greatly improved by simply increasing the rotating rate from 500 to 1,500 r/min as shown in our original manuscript.

Similar to the GO/rGO films prepared by other methods, the adjacent layers are attracted each other mainly by Van der Waal's force. The van der Waals force can be estimated by Lifshitz's formulation $F = S \frac{A_{\text{Ham}}}{6\pi d^3}$, where A_{Ham} is the Hamaker coefficient, d and S are the interlayer distance and contact area of the adjacent GO/rGO nanosheets. It can be seen from this formula that the binding force is highly sensitive to the interlayer distance and proportional to the contact area between

adjacent GO/rGO layers. For GO-based films, the position of XRD peak (θ) determines the interlayer distance (d), while the FWHM reflects the ordering, which is correlated with the alignment degree. Normally, the better alignment can increase the contact and interaction. As shown in our manuscript, the XRD peak upshifts along with a great decrease in FWHM with increasing the rotation rate. This indicates that the connection and van der Waals force between stacked adjacent nanosheets are improved. As a consequence, the mechanical strength of GO films as well as both the mechanical strength and electrical conductivity of the rGO films increases with increasing the rotating rate. As shown in Supplementary Table 1 and Table R1, the GO/rGO films achieved by our method at high rotating rates show better alignment and higher compaction, which are responsible for the better properties than those reported in the literature.

In our original manuscript, we have discussed the origin of the high performance of the supercapacitors made by highly aligned and compact rGO/SWCNT hybrid films as follows. It is well known that SCs store and release electrical energy based on the electrostatic interactions between ions in the electrolyte and electrodes near the electrode surface. At micro-scale, the well-aligned rGO and SWCNTs in the hybrid film form parallelly oriented interlayer galleries, which effectively restrict the electrolyte in such 2D space and consequently could facilitate the fast transport of ions. More importantly, the high specific surface area and the compact layered structure of rGO/SWCNT electrodes not only allow the penetration of adequate amount of electrolyte but also greatly reduce the amount of in-effective electrolyte in the interlayer galleries, which ensure high volumetric and gravimetric capacitances. At macro-scale, because of the planar electrodes layout, ions can transport fast between the two electrodes reversibly without through considerable tortuosity as the case of graphene fiber. In addition, the good alignment and compaction lead to high electrical conductivity, good flexibility and high tensile strength. The good electrical conductivity can facilitate the electron transport in electrodes, and the high tensile strength enables their use as flexible electrodes. Therefore, the rGO/SWCNT hybrid

films show excellent performances for SC applications.

In the revised manuscript, we have added the above discussions to explain why the centrifugal alignment results in improved performance in different applications.

The rotating rate is apparently a very important parameter to a successfully aligned film. It effects shear stress and centrifugal force. An increased rotating rate could make the film more compact and aligned. But a higher rotating rate could lead to a turbulent flow which damage the alignment of GO nanosheets. So it's recommended that the authors provide an empirical equation that determines the maximum rotating rate considering the factors including solution viscosity, RHT diameter.

Reply: We thank the reviewer very much for valuable suggestion.

Based on the fluid mechanics, there are mainly two forces that control the behavior of the flow, inertial force and viscose force. The former can be considered as the dynamic sensitivity of flow, while the latter as the dissipation capability for the disturbance. Typically, the relative contribution of these two forces can be estimated by the Reynolds number, which is the ratio of inertial force to viscose force. In our study, the liquid layer around the inside surface of a rotating hollow cylinder, typically considered as rimming flow, is a non-trivial example of a steady, two-dimensional, viscous flow with a free surface [*J. Fluid. Mech.* 190, 321-342 (1988); *J. Fluid. Mech.* 84, 145-165 (1978)]. The Reynolds number is expressed as:

$$Re \equiv \omega D^2 / \nu$$

Where ω , D and ν are the angular velocity of the rotating tube and thickness of the and kinematic viscosity of the flow, respectively.

When the Reynolds number is small, any disturbance, including the change of the tube rotation rate and any uncertainty of the flow boundary condition (such as the roughness of the inner hollow tube), can be dissipated and the flow exhibits characteristic of laminar flow. When the Reynolds number is large, any disturbance will be amplified and eventually results in the generation of eddies, and the flow

exhibits characteristic of turbulence. In fact, the turbulence flow is extensively considered as the mechanism for dispersion and mixing. Therefore, the flow should be controlled within the laminar flow in order to align the 2D sheets.

In order to guarantee the liquid flow is located in the region of laminar flow, Reynolds number must be depressed, that is, viscose effects dominate the behavior of the flow. The critical Reynolds number for the transition between laminar flow and turbulent flow is dependent on the specific physical properties of the flow. Here, we take the critical Reynolds number to be one, to roughly estimate the domains corresponding to laminar flow phase and turbulent phase, and the phase diagram obtained is shown in Fig. R3. Note that the viscosity of GO dispersion is increased along with the evaporation of water induced by heating during the continuous centrifugal casting process. As shown in Fig. R3, the GO flow is always limited in the laminar flow region at all the rotating rates used in our experiments even though the viscosity increase is not considered. According to this phase diagram, the maximum rotating rate that limits the solution in the laminar flow region should be no less than 10^9 rad/s, corresponding to tube rotating rate of $\sim 10^{10}$ r/min, considering the viscosity of the GO dispersion used. This means the tube rotating rate can be further increased to improve the properties of the films. For instance, we have synthesized GO films by increasing the rotating rate to 2,500 r/min while keeping the other parameters unchanged. The GO films obtained show better alignment and compaction than those synthesized at 1,500 r/min (Fig. R2), and the reduced GO films also show much higher tensile strength of ~ 660 MPa and electrical conductivity of ~ 650 S/cm.

We have added the above discussions and the related data in the revised manuscript.

The authors demonstrated a smaller interlayer spacing and FWHM of GO obtained at 1500 r/min, as compared to those of the reported GO films. Though the higher rotating rate could improve the compact of GO films (Fig 1e), the authors should acknowledge the possible contribution of heating (80 C in this work) to the smaller interlayer spacing. Heating could remove the residue water often seen in other

methods like vacuum filtration and solvent evaporation, and even reduce GO films to some extent.

Reply: We thank the reviewer very much for kind suggestion. Indeed, as the reviewer mentioned, the removal of residual water between GO nanosheets during heating can reduce the interlayer spacing, which consequently helps the compaction of GO films. Because of the very short synthesis time (for instance, 1 min for 10 μm -thick film) and low heating temperature, no reduction was observed in our case. In addition, during heating, the viscosity of the GO dispersion is increased as well, which leads to an increase in shear stress according to the equation, $\sigma_{\text{shear}} = \mu \omega e_{\theta}$, where μ , ω , and e_{θ} are the viscosity of the liquid and the angular velocity and unit tangent vector of the rotating tube, respectively. This would be beneficial for aligning the GO nanosheets.

We have acknowledged the contribution of heating to the alignment and compaction of GO films in the revised manuscript.

The authors should show the demonstration of nanoscale-thick film, which is appealing and has potential applications in a wide range of applications. The authors mentioned this method is applicable to make film with thickness from a few nanometers to hundreds of micrometers, but only demonstrated the formation and various applications of microscale-thick GO/rGO films, and the claim on forming nanometer thick film is made without experimental proof. As a matter of fact, it is often more challenging and desired to prepare a uniform nanoscale-thick film.

Reply: We thank the reviewer very much for kind suggestion. We have synthesized nanoscale-thick GO films on PET and 4 inch silicon by continuous centrifugal casting method using a dilute GO dispersion (0.05 mg/mL) at a rotating rate of 1,500 r/min (Fig. R10a,b). AFM image shows that the film has a smooth surface and a uniform thickness of ~ 50 nm (Fig. R10c,d).

We have added these data in the revised manuscript.

Figure R10. (a) Photo of a meter-scale thin GO film on PET produced with a rotating tube of 33 cm in inner diameter. (b) A thin GO film on a 4 inch silicon wafer prepared with the same conditions as those for the production of the film in **a**. (c) AFM image of the film in **b** and (d) the height profiles along the three lines in **c**, showing the thickness of ~50 nm.

Line 130, "flexibility" should be changed to "flexible"

Reply: We have changed the word “flexibility” to “flexible” in the revised manuscript.

REVIEWERS' COMMENTS:

Reviewer #1 (Remarks to the Author):

1. The authors say that residual SDS and HI resulted in the deviations in CV curve shapes and low Coulombic efficiencies of the GCD curves in the original MS. They indicate removing residual SDS and HI almost entirely after 18 h washing. In the revised MS the authors provide FTIR and XPS results supporting their substantial removal. What remains unclear is: How do highly conductive films cause resistive CV curves? A detailed EIS analysis may help to explain this?

A parametric study of rG-O/SWCNT films with different thicknesses for fabricated supercapacitor devices is likely to make the reported work more useful.

For Figure R8 and Figure 4e, there might be a mistake and it would seem that "square" and "star" signs should be exchanged for "device" and "electrode" signs.

4) There are "typo" mistakes in the revised MS. There should not be.

5) In the Experimental section, the SWCNT concentration should be provided.

Reviewer #2 (Remarks to the Author):

I'm satisfied with the changes introduced by the authors and recommend the revised manuscript for publication.

Reviewer #3 (Remarks to the Author):

The authors have satisfactorily addressed my comments.

Reviewer #4 (Remarks to the Author):

I have read through revised paper and the response of the authors and support the publication of this work in Nature Communications.

The important finding in this work, which deserves to be disseminated to the community, is the demonstration that continuous centrifugal casting method is a very efficient way to fabricate highly compact and well aligned films composed of stacked and overlapped GO nanosheets, which can be readily scale up to make commercially relevant filtration and energy storage membranes. Most significantly, the fabrication of well aligned membrane can be obtained in significantly much shorter time than those made from vacuum filtration (many orders of improvement in terms of time !!). This advancement should not be trivialized as the progress in graphene research requires such breakthrough for translation to industrial applications.

Besides the engineering advancement, the authors have provided sufficient scientific explanations of the fluid mechanics involved in the alignment. For example they explained that during the continuous centrifugal casting process, the G-O nanosheets are subjected to shear stress and centrifugal stress at the same time; the shear stress is along the tangential direction of the rotating tube, which not only can align but also can smooth the G-O nanosheets, while the centrifugal stress is along the radial direction of the rotating tube, which provide the force for condense stacking of parallel aligned smooth G-O nanosheets. The referee feels that this method is also applicable to other 2D materials synthesized by solution-exfoliation methods and allows composites 2D films to be fabricated easily.

The authors have also shown that using a variety of characterization as well as proof of concept applications, the formation mechanism of such highly aligned and compact films leads to improved performance in supercapacitors.

This method of forming graphene oxide membrane will certainly be adopted by both the research community and industry in the future due to its efficiency and effectiveness, thus I support the immediate publication of this work in Nature Communications.

Response to reviewer's comments

Reviewer #1 (Remarks to the Author):

1. The authors say that residual SDS and HI resulted in the deviations in CV curve shapes and low Coulombic efficiencies of the GCD curves in the original MS. They indicate removing residual SDS and HI almost entirely after 18 h washing. In the revised MS the authors provide FTIR and XPS results supporting their substantial removal. What remains unclear is: How do highly conductive films cause resistive CV curves? A detailed EIS analysis may help to explain this?

Response: We thank the reviewer very much for the comments.

As we know, the supercapacitor performance is not only determined by the properties of electrode materials, but also the geometry of the electrode layout and the distance of both electrodes. For typical coin-shaped supercapacitors, as reported by Prof. Ruoff in 2008, the electron mainly transports along the thickness direction of the electrodes (Figure R1a). In contrast, for our ribbon supercapacitors and the reported fiber supercapacitors (For instance, the fibrous supercapacitor reported by Prof. Yuan Chen and Prof. Liming Dai, scalable synthesis of hierarchically structured carbon nanotube-graphene fibres for capacitive energy storage. *Nat. Nanotech.* 9, 555-562, 2014), the electrons mainly transport along the axial direction of the electrode, as indicated in the Figure R1b. Such geometrical difference results in huge difference in the electrode resistance, for a given electrode material, which can be estimated as follows. Assuming that the electrodes have the geometries as shown in Figure R1a, b, the resistance of the electrode in ribbon supercapacitor is significantly higher than that in coin-shaped supercapacitor by six orders of magnitude. This means the high conductivity of our highly conductive graphene/CNT hybrid electrode can only partially compensate the high resistance caused by the electrode layout, which is the reason for the resistive CV curves observed in our ribbon supercapacitors. Similarly, because of such geometrical reasons, nearly all the reported fiber supercapacitor exhibited resistive CV curves, especially when the length of fiber electrode is large.

We have added the following sentence to explain the resistive CV curves in the revised manuscript: The observed resistive CV curves might be due to the specific electrode layout of tape SCs and the resultant large electrode resistance, similar to the reported fiber SCs^[41].

Figure R1. Schematic for the electron transport in the (a) coin-shaped supercapacitor and (b) ribbon supercapacitor.

A parametric study of rG-O/SWCNT films with different thicknesses for fabricated supercapacitor devices is likely to make the reported work more useful.

Response: We thank the review very much for kind suggestion.

It has been reported by many groups that the performance of electrode is sensitive to the thickness of electrode, when it is in the range of nanometer thickness. Usually, a small thickness results in high capacitance values. In order to avoid such exaggeration, it was suggested by Prof. Ruoff that the thickness of the electrode is preferred to be larger than 10 μm (*Energy Environ. Sci.* 3, 1294-1301, 2010). It is worth noting that the thickness of our electrode is $\sim 39 \mu\text{m}$, which is already comparable to the thickness of commercially available supercapacitors. We have also tested the capacitive performance of the electrodes with thickness of $\sim 12 \mu\text{m}$ and $\sim 23 \mu\text{m}$, and the volumetric capacitance obtained are 433 F/cm^3 and 419 F/cm^3 , respectively, at a current density of $\sim 110 \text{ mA/cm}^2$. This indicates that at the tens of micrometer scale, the thickness of the electrode has a small influence on the performance of supercapacitors.

We have added the following discussion in the revised manuscript: At the tens of micrometer scale, the thickness of the electrode has a small influence on the performance of the SCs. For example, reducing the thickness of electrodes to ~ 23 and $12 \mu\text{m}$ only leads to a small increase in volumetric capacitance to 419 and 433 F cm^{-3} , respectively, at a current density of $\sim 110 \text{ mA cm}^{-2}$.

For Figure R8 and Figure 4e, there might be a mistake and it would seem that “square” and “star” signs should be exchanged for “device” and “electrode” signs.

Response: We thank the reviewer very much for kind reminder. We have exchanged the

signs in the revised manuscript.

There are “typo” mistakes in the revised MS. There should not be.

Response: We have carefully checked the manuscript to remove the “typo” mistakes.

In the experimental section, the SWCNT concentration should be provided.

Response: According to the reviewer’s kind suggestion, we have added the SWCNT concentration (5 wt%) in the revised manuscript.

Reviewer #2 (Remarks to the Author):

I'm satisfied with the changes introduced by the authors and recommend the revised manuscript for publication.

Reviewer #3 (Remarks to the Author):

The authors have satisfactorily addressed my comments.

Reviewer #4 (Remarks to the Author):

I have read through revised paper and the response of the authors and support the publication of this work in Nature Communications.

The important finding in this work, which deserves to be disseminated to the community, is the demonstration that continuous centrifugal casting method is a very efficient way to fabricate highly compact and well aligned films composed of stacked and overlapped GO nanosheets, which can be readily scale up to make commercially relevant filtration and energy storage membranes. Most significantly, the fabrication of well aligned membrane can be obtained in significantly much shorter time than those made from vacuum filtration (many orders of improvement in terms of time !!). This advancement should not be trivialized as the progress in graphene research requires such breakthrough for translation to industrial applications.

Besides the engineering advancement, the authors have provided sufficient scientific explanations of the fluid mechanics involved in the alignment. For example they explained that during the continuous centrifugal casting process, the G-O nanosheets are subjected to shear stress and centrifugal stress at the same time; the shear stress is along the tangential direction of the rotating tube, which not only can align but also can smooth the G-O nanosheets, while the centrifugal stress is along the radial direction of the rotating tube, which provide the force for condense stacking of parallel aligned smooth G-O nanosheets. The referee feels that this method is also applicable to other 2D materials synthesized by solution-exfoliation methods and allows composites 2D films to be fabricated easily.

The authors have also shown that using a variety of characterization as well as proof of concept applications, the formation mechanism of such highly aligned and compact films leads to improved performance in supercapacitors.

This method of forming graphene oxide membrane will certainly be adopted by both the research community and industry in the future due to its efficiency and effectiveness, thus I support the immediate publication of this work in Nature Communications.